# A synthetic cell with integrated DNA self-replication and lipid biosynthesis

Ana María Restrepo Sierra[1], Federico Ramirez Gomez [®][1], Mats van Tongeren[1], Laura Sierra Heras [®][2] & Christophe Danelon [®][1,2] [✉]

The emergence, organization, and persistence of cellular life are the result of the functional integration of metabolic and genetic networks. Here, we engineer phospholipid vesicles that can operate three essential functions, namely transcription-translation of a partial genome, self-replication of this DNA program, and membrane synthesis. The synthetic genome encodes six proteins, and its compartmentalized expression produces active liposomes with distinct phenotypes demonstrating successful module integration. Our results reveal that genetic factors exert a stronger control over DNA replication and membrane synthesis than metabolic crosstalk or module co-activity. By showing how genetically encoded functions derived from different species can be integrated in liposome compartments, our work opens avenues for the construction of autonomous and evolving synthetic cells.

The construction of a synthetic cell from the bottom up is a grand challenge at the intersection of bioscience and engineering. Inspired by the observation of common processes in all living organisms, researchers have started to build some of the essential cellular functions, hereafter called 'modules', in vitro. The expanding repertoire of genetic parts and characterized biochemical networks has enabled the cell-free reconstitution of life's fundamental mechanisms, such as the synthesis of membrane constituents[1–3], division related processes[4,5], DNA replication[6,7], energy regeneration[8], and cell–cell communication[9]. While these studies have yielded valuable insights into the specifics of each biological module, they have not addressed the higher-ordered complexity that lies in the integration of multiple processes, in particular when the involved genetic or protein parts are derived from various organisms[10,11].

Three subsystems appear essential for basic cellular life: a vesicular system defining an internal machinery that synthesizes its membrane constituents, a replicable template that carries information, and a metabolic cycle that produces the molecular components[12,13]. As a construction paradigm, we envisioned that in vitro transcription-translation (IVTT) of a synthetic DNA template using recombinant elements (PURE system)[14] inside phospholipid vesicles (liposomes) would constitute the scaffold onto which biological functions can be implemented to create an autonomously living synthetic cell. In contrast to other approaches which rely exclusively on purified proteins or cell lysates, our DNA-based architecture enables replication, system's level evolution, and is constituted of well-defined components[11]. Replication of the DNA program can be seen as the seed module priming self-maintenance and evolvability[15]. However, an experimental demonstration of module integration directed by a synthetic self-replicating genome has remained elusive.

Here, we pin the work for combining synthetic cell modules by constructing a self-replicating DNA genome, named *DNArep-PLsyn*, encoding both a DNA replication machinery (DNArep) and a phospholipid biosynthesis pathway (PLsyn). We establish the conditions for in-liposome expression of *DNArep-PLsyn* with PURE system and demonstrate the combined activities of universal cellular modules in a minimal in vitro system.

## Results

### Design and cell-free expression of a synthetic replicating genome

We constructed a synthetic DNA replication system following the design of the Φ29 genome[16] which consists of a linear DNA template with origin of replication sequences at each end. Previous work showed that four phage proteins—the terminal protein (TP) that functions as a replication primer, the DNA polymerase (DNAP), the

[1]Department of Bionanoscience, Kavli Institute of Nanoscience, Delft University of Technology, Delft, The Netherlands. [2]Toulouse Biotechnology Institute (TBI), Université de Toulouse, CNRS, INRAE, INSA, Toulouse, France. [✉]e-mail: danelon@insa-toulouse.fr

single-stranded DNA binding protein (SSB), and the double-stranded DNA binding protein (DSB)—were sufficient to replicate a linear DNA in vitro[17]. Moreover, we previously showed that expression in PURE system of a minimal φ29-based linear replicon encoding DNAP (*p2* gene) and TP (*p3* gene) led to exponential amplification of DNA, also when the reaction was compartmentalized inside micrometer-sized liposomes[7,15]. We here sought to integrate additional genes into this seed replication module and hypothesized that the larger synthetic genome could be replicated—and all the gene products could be synthesized—upon expression in PURE system (Fig. 1a–d). The newly introduced genes encode four enzymes of the *E. coli* Kennedy pathway: *sn*-Glycerol-3-phosphate acyltransferase (PlsB), Lysophosphatidic acid acyltransferase (PlsC), Phosphatidate cytidylyltransferase (CdsA), and Phosphatidylserine synthase (PssA) (Fig. 1d). These enzymes catalyze the sequential conversion of oleoyl-CoA and glycerol-3-phosphate precursors into 1,2-dioleoyl-*sn*-glycero-3-phospho-L-serine (DOPS), the last intermediate for 1,2-dioleoyl-*sn*-glycero-3-phosphoethanolamine (DOPE) production. Membrane synthesis in gene-expressing vesicles can then be visualized using a PS-specific fluorescent probe[2]. Therefore, our final linear genome, named *DNArep-PLsyn*, is flanked with φ29 origins of replication on each end, and it encompasses six genes (two

for *DNArep* and four for *PLsyn*) as individual transcription units (Fig. 1a).

To construct the *DNArep-PLsyn* synthetic genome, we iterated throughout different cloning strategies and found that template complexity (i.e., repetitive elements) often led to recombination events in *E. coli*. We then opted for an in vitro DNA assembly approach using overlapping polymerase chain reaction (PCR) to stitch the *DNArep* and *PLsyn* genetic parts (Supplementary Fig. 1), and a yeast-based cloning approach (Supplementary Fig. 2). Notably, *S. cerevisiae* yeast did not seem to pose recombination issues with repetitive regulatory sequences, unlike *E. coli*. After plasmid extraction from yeast, we generated the linear *DNArep-PLsyn* genome by PCR. In both in vitro and in-yeast DNA assemblies, we successfully obtained a linear template with the expected size (~9600 bp), and the sequence was validated by nanopore sequencing (Supplementary Fig. 1 and 2).

Next, we confirmed that the expression of *DNArep-PLsyn* with PURE system generates the six encoded proteins. The reaction mix was supplemented with GreenLys reagent for co-translational protein labeling. All six proteins were produced at detectable levels starting from DNA assembled in vitro or in yeast (Fig. 1e, Supplementary Fig. 3 and 4). Interestingly, only a slight reduction of protein expres-

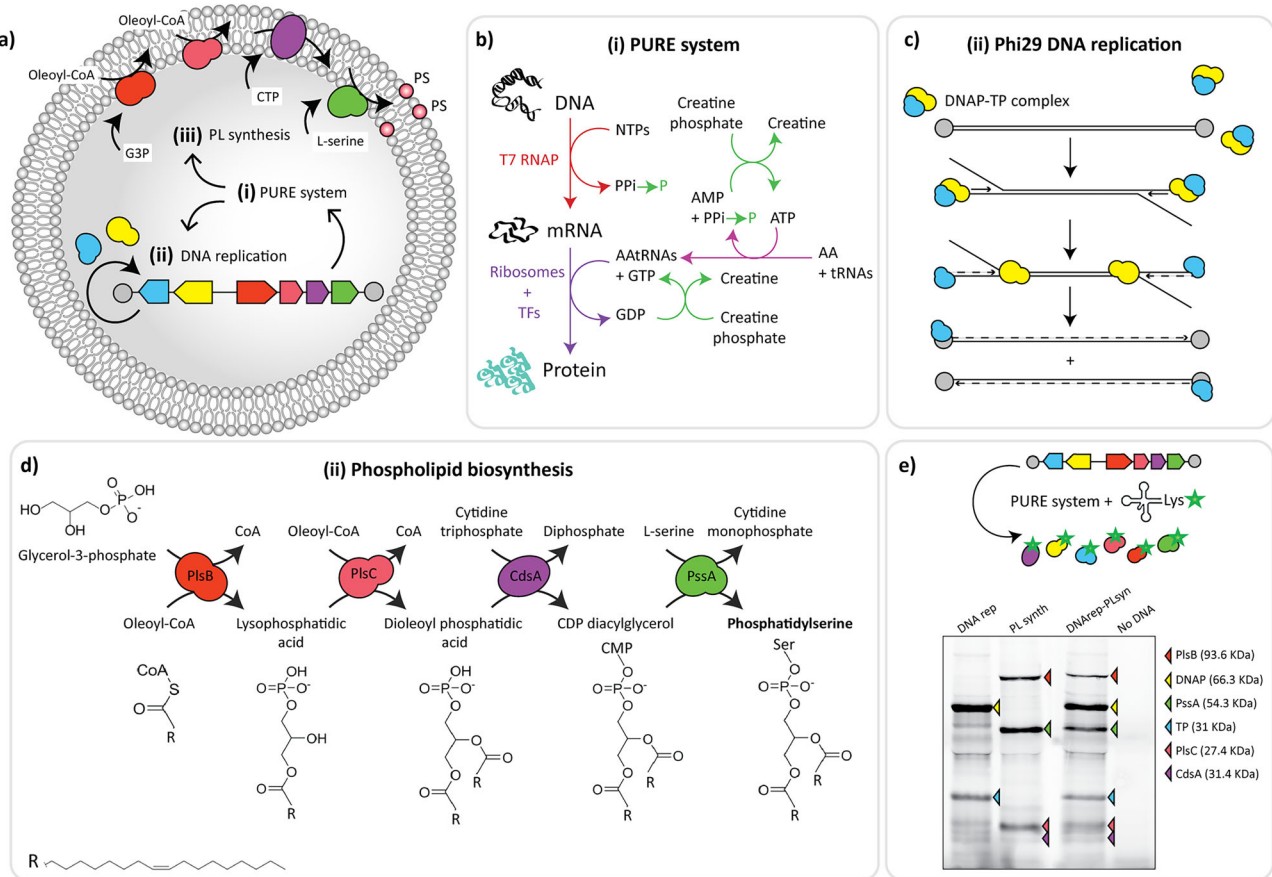

**Fig. 1 | A synthetic genome encoding two cellular modules. a** Synthetic vesicles with encapsulated *DNArep-PLsyn* genome and coupled transcription-translation, DNA self-replication, and phospholipid biosynthesis. **b** PURE system served as the main metabolic machinery for transcription and translation of DNA-encoded proteins with a creatine phosphate-based energy regeneration system. **c** Initiation and elongation steps of the protein-primed DNA replication mechanism from the bacteriophage φ29. The dashed lines depict the newly synthesized strands. **d** A four enzyme-cascade of the *E. coli* Kennedy pathway transforms oleoyl-CoA and glycerol 3-phosphate (G3P) into dioleoyl-phosphatidylserine (PS). **e** SDS-PAGE analysis of bulk IVTT reactions from the assembled *DNArep-PLsyn* template, or from the

individual *DNArep* and *PLsyn* fragments. The PURE system solution was supplemented with GreenLys reagent for fluorescent labeling of the synthesized proteins (indicated with arrowheads). The uncropped gel with ladder lane and replicates can be found in Supplementary Fig. 3. Other abbreviations: AA amino acid, AAtRNA amino acid-loaded tRNA, AMP adenosine monophosphate, ATP adenosine triphosphate, CoA coenzyme A, CDP cytosine diphosphate, CMP cytosine monophosphate, CTP cytosine triphosphate, GDP guanosine diphosphate, GTP guanosine triphosphate, NTP nucleoside triphosphate, PPi pyrophosphate, TF translation factor. Source data are available for this figure in the Source Data file.

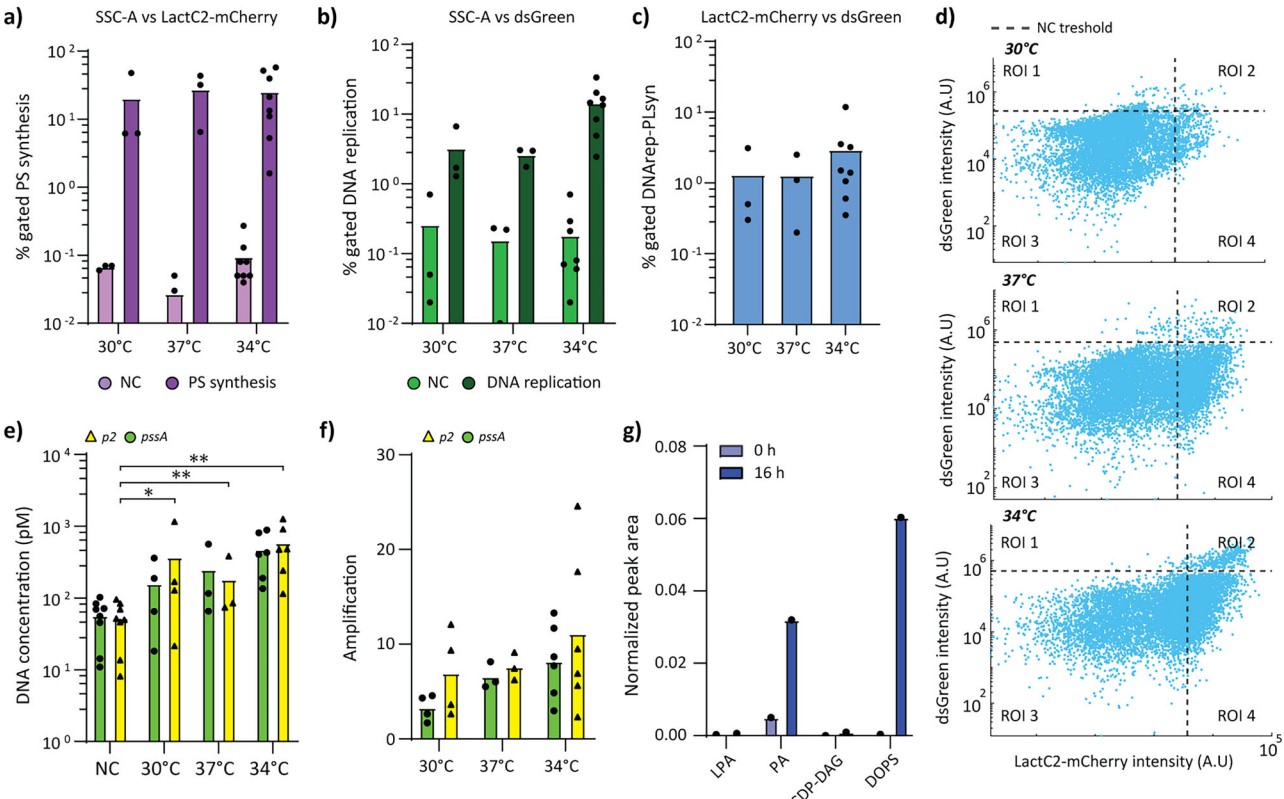

**Fig. 2 | Validation of DNArep and PLsyn protein activity inside gene-expressing liposomes at different incubation temperatures.** Percentage of liposomes with active DOPS synthesis (**a**) and active DNA replication (**b**) under 30 °C, 37 °C and 34 °C incubation temperatures. Flow cytometry data are SSC-A vs. dsGreen for DNA replication and SSC-A vs. LactC2-mCherry for DOPS synthesis. Data points represent biological repeats and bar height the mean value. Raw data from individual replicates can be found in Supplementary Fig. 5. NC refers to samples, where DNA was omitted but the solutions were incubated at the indicated temperature. **c** Percentage of liposomes exhibiting dual dsGreen and LactC2-mCherry signals at 30 °C, 37 °C, and 34 °C incubation temperatures. Joint phenotype populations were selected from LactC2-mCherry vs dsGreen scatter plots. Raw data from individual replicates can be found in Supplementary Fig. 5. **d** Flow cytometry scatter plots from liposome samples displaying four regions of interest (ROI 1-4) at all tested temperatures: DNArep-active liposomes are in ROI 1, PLsyn-active liposomes in ROI 4, and liposomes with both active DNArep and PLsyn modules are in ROI 2. Vertical and horizontal dashed lines indicate intensity threshold values that have been defined using control samples (see Supplementary Fig. 5). Data from additional biological repeats can be found in Supplementary Fig. 5. **e** Absolute DNA

quantification by qPCR of samples incubated at 30 °C, 37 °C, and 34 °C. qPCR target regions (~200 bp) are from *pssA* and *p2* genes. The negative control (NC) represents calculated DNA values at initial incubation points (0 h). DNA concentration changes between 0 h (NC) and 16 h were assessed using a two-sided log-ratio paired *t*-test. Log-transformed ratios of 16-h to 0-h values were calculated for each replicate and each corresponding gene (*pssA* and *p2*). A one-sample *t*-test was then performed to determine if the mean log-ratio significantly differed from zero ($p < 0.05$). * $p \leq 0.05$, ** $p \leq 0.01$, and *** $p \leq 0.001$. Exact *p* values for the genes *p2/pssA* are: 0.0173/0.0178 (30 °C), 0.003/0.00397 (37 °C), and 0.00162/0.00035 (34 °C). **f** Amplification fold of *DNArep-PLsyn* DNA calculated from qPCR data in panel e: end-point (16 h) DNA concentration / DNA concentration at time zero. Data points represent biological repeats and bar height the mean value. **g** LC-MS detection of DOPS and PLsyn intermediate enzymatic products before and after expression of the *DNArep-PLsyn* genome. Peak area for each compound was normalized to that of DOPG. Additional biological repeats and negative controls can be found in Supplementary Fig. 8. Source data are available for this figure in the Source Data file.

sion levels was observed for the *DNArep-PLsyn* template compared to the separate expression of each individual genetic module (Fig. 1e and Supplementary Fig. 3). This could be caused by resource sharing when the number of genes increases, but the effect was less pronounced than expected. We conclude that *DNArep-PLsyn* acts as an effective template for expressing all necessary proteins involved in both DNArep and PLsyn modules.

**Integration of DNArep and PLsyn modules inside gene-expressing vesicles**

Our next aim was to evaluate and potentially optimize the simultaneous activity of the DNArep and PLsyn modules inside liposomes. Since each of the encoded modules may have a preferred reaction temperature (DNA replication works well at ~30 °C[7,17], while cell-free gene expression[18] and phospholipid biosynthesis[2] are most effective at 37 °C), we decided to test different incubation temperatures. We encapsulated in liposomes the *DNArep-PLsyn* genome together with

PURE system and the required substrates and cofactors for both DNArep and Plsyn, and we ran the reactions at 30 °C, 34 °C, or 37 °C. After overnight incubation, we stained the DNA with the dsGreen intercalating dye[19] and the membrane-incorporated DOPS with the PS-specific probe LactC2-mCherry[2], and we analyzed the samples by flow cytometry (Fig. 2a–d). For each fluorescent probe, we performed an intensity thresholding based on negative control samples (Supplementary Fig. 5), thus defining four regions of interest (ROI) in the scatter plot (Fig. 2d). Liposomes exhibiting functional DNArep (ROI 1 + ROI 2) or PLsyn (ROI 2 + ROI 4) modules were detected at all three temperatures (Fig. 2a, b), with a slightly higher occurrence for DNArep-active liposomes at 34 °C than at 30 and 37 °C (Fig. 2b–d). Notably, a range of 0.4 to 12% of the liposomes (corresponding to ~50–1200 liposomes per sample across biological replicates at 34 °C) localized in ROI 2 indicating that both DNArep and PLsyn modules were simultaneously active (Fig. 2c, d, and Supplementary Fig. 5). A larger fraction of liposomes was positive to either one of the two modules (ROI 1 or

ROI 4) or was inactive (ROI 3) (Fig. 2a, b, d). Such a heterogeneity within the same clonal (here referring to the fact that one DNA species was used) population of liposomes is also observed in single-gene expression experiments and can be attributed to uneven loading or supply of substrates or cofactors, or to varying expression levels of the genetic modules between liposomes[20]. In addition, a significant variability across biological replicates (sample-to-sample heterogeneity) was observed. For example, the percentage of DNArep-PLsyn-positive liposomes (ROI 2) was 2.9 ± 1.3% (mean ± s.e.m) across eight biological replicates at 34 °C (Supplementary Fig. 5). Nonetheless, these results demonstrate that functional integration of DNArep and PLsyn modules from a synthetic genome is possible at temperatures ranging from 30 °C to 37 °C.

To provide more direct evidence of genome self-replication, we measured the concentration of DNA using quantitative PCR (qPCR). Two different sequences localized in opposite regions of the linear *DNArep-PLsyn* genome were targeted for qPCR, one in the *p2* gene and one in the *pssA* gene. The results quantitatively confirmed that all tested temperatures supported genome replication, again with a slight preference for 34 °C (Fig. 2e, f). We further investigated whether the full-length genome was amplified (vs. shorter amplicons) by targeting all six genes by qPCR. These experiments were performed at 34 °C. Despite some variations in the concentration of replicated genes, the data showed that the entire DNA sequence between the *p3* and *pssA* genes (~5000 bp apart) was amplified about 10-fold (Supplementary Fig. 6). Small differences could arise from DNA replication arrest events, leading to incomplete fragment amplification[21], or from qPCR-related variations in the gene-specific primer design and efficiency. Since qPCR amplifies only ~200-bp regions and the terminal origins of replication were not targeted, we also recovered DNA from liposome samples by PCR followed by agarose gel analysis of the amplification products. The entire *DNArep-PLsyn* genome (within the resolution of agarose gel electrophoresis) could be recovered from diluted liposome samples (Supplementary Figs. 6 and 7). Shorter DNA species were also observed, suggesting that the *DNArep-PLsyn* genome may have experienced incomplete self-replication or that smaller DNA fragments were generated during PCR recovery.

Next, we sought to directly demonstrate the production of PS and intermediate enzymatic products of the reconstituted phospholipid biosynthesis pathway by liquid chromatography-mass spectrometry (LC-MS). We found that DOPS was produced, although not in high concentrations (Fig. 2g and Supplementary Fig. 8). Moreover, DOPA was accumulated, suggesting that CdsA may be limiting the yield of PS production. Considering that dioleoyl-phosphatidylglycerol (DOPG) accounts for 12% of the total lipids, we estimated that synthesized DOPS would represent 0.7% of the total lipid content after 16 h incubation at 34 °C. However, it is relevant to note that LC-MS gives ensemble measurements, the obtained concentration values reflecting the average activity of all the liposomes in the sample. Individual vesicles may contain none or higher-than-average amounts of DOPS (see next section).

## High-content imaging of DNArep and PLsyn phenotypes

Having established the successful integration of the DNArep and PLsyn modules, we aimed to directly visualize the different liposome phenotypes, allowing for a more accurate classification based on activity levels. In particular, we asked whether liposome size, lamellarity, or morphology could affect or be affected by module activity. We combined fluorescence confocal microscopy with an in-house-developed software called SMELDit to enable automated liposome identification, feature analysis, and image recovery from scattered data plots (see Methods). We expressed *DNArep-PLsyn* in liposomes at 34 °C and used the dsGreen and LactC2-mCherry signals as fluorescent markers for DNArep and PLsyn activity, respectively (Fig. 3a). We observed that the addition of the substrates and cofactors, and

the expression of *DNArep-PLsyn* did not affect liposome sample quality (Fig. 3a, Supplementary Fig. 9, and Supplementary Movie 1). Notably, images unraveled phenotypic traits that could not be inferred from flow cytometry data, such as the presence of bright dsGreen spots in the vesicle lumen, which result from active DNA replication. We had already observed a similar phenotype during amplification of a shorter DNA self-replicator, which was attributed to an induced condensation of highly concentrated DNA[7,22]. Here, it is interesting to see that such a phenomenon is also possible with a 3-fold longer DNA template (~9.6 kb vs. ~3.2 kb) containing more expressed genes (6 vs. 2).

When aggregating data from all biological replicates, over 34,000 liposomes were recognized. We generated a phenotype map corresponding to the two-dimensional plot of LactC2-mCherry vs. dsGreen signals from single vesicles (Fig. 3b). Liposomes were classified according to four different phenotypes based on intensity thresholding, akin to flow cytometry data analysis (ROI 1-4) (Fig. 2d). We found that ~8% of liposomes, corresponding to over 2900 liposomes, had coexisting DNA replication and DOPS synthesis (ROI 2). Vesicles with either active PLsyn (~10%, ROI 4) or active DNArep (~31%, ROI 1) module were more abundant (Fig. 3b). We then questioned whether liposome sizes varied across the four regions, for example as a result of membrane synthesis. Vesicle size distribution was computed for each phenotypic region (Supplementary Fig. 10). We observed no marked differences in the median values of the apparent diameter between active (ROI 2 and 4) and inactive (ROI 1 and 3) PLsyn module (3.7 ± 2.4 μm median across all ROIs), indicating that the yield of newly synthesized lipids is not sufficient for detectable physical growth of liposomes.

To account for the variability across biological replicates, we constructed the phenotype map for each replicate sample (Fig. 3c and Supplementary Fig. 11). Despite clear variations in the percentages of gated liposomes in each region, all replicates contained vesicles exhibiting simultaneous DNArep and PLsyn activity (Fig. 3c). Finally, we examined the LactC2-mCherry and dsGreen intensity values for every liposome as this may reveal differences in the efficacy of a given module when operating alone or together. From both pooled data and individual replicates, we observed no strong differences in the intensity pattern of the module activity reporter dyes between ROIs (Fig. 3d). This result suggests that DNArep activity is not lessened when coupled with PLsyn activity, and vice versa. These findings point to a robust compatibility between the two functions. In addition, some liposomes exhibit an intensity of the DOPS probe that can be over one order of magnitude higher than the average value (Fig. 3d), indicating that synthesized DOPS could represent up to 7% (0.7 × 10) of the total lipid content.

For better generalization of image analysis across experiments and sample types, we developed a deep-learning-based method for automated vesicle segmentation and phenotype classification. The model was able to identify 306 liposomes with 90% confidence out of 353 manually identified vesicles (Supplementary Fig. 13). Moreover, the model enabled classification of the phenotypes, including vesicles with localized replication foci, without relying on fluorescence intensity thresholding or other predefined parameters (Supplementary Fig. 14).

## Metabolic and activity crosstalk between DNArep and PLsyn modules

To better understand the influence that active DNArep or PLsyn modules may have on each other, we assayed liposomes expressing the full synthetic genome, this time by adding either of the two sets of substrates/cofactors (DNArep or PLsyn) (Fig. 4a). An additional condition was tested, where all DNArep substrates/cofactors were supplied, except for dNTPs. This switches DNArep module OFF but allows to study the effect of the other molecules (i.e., SSB, DSB, ammonium

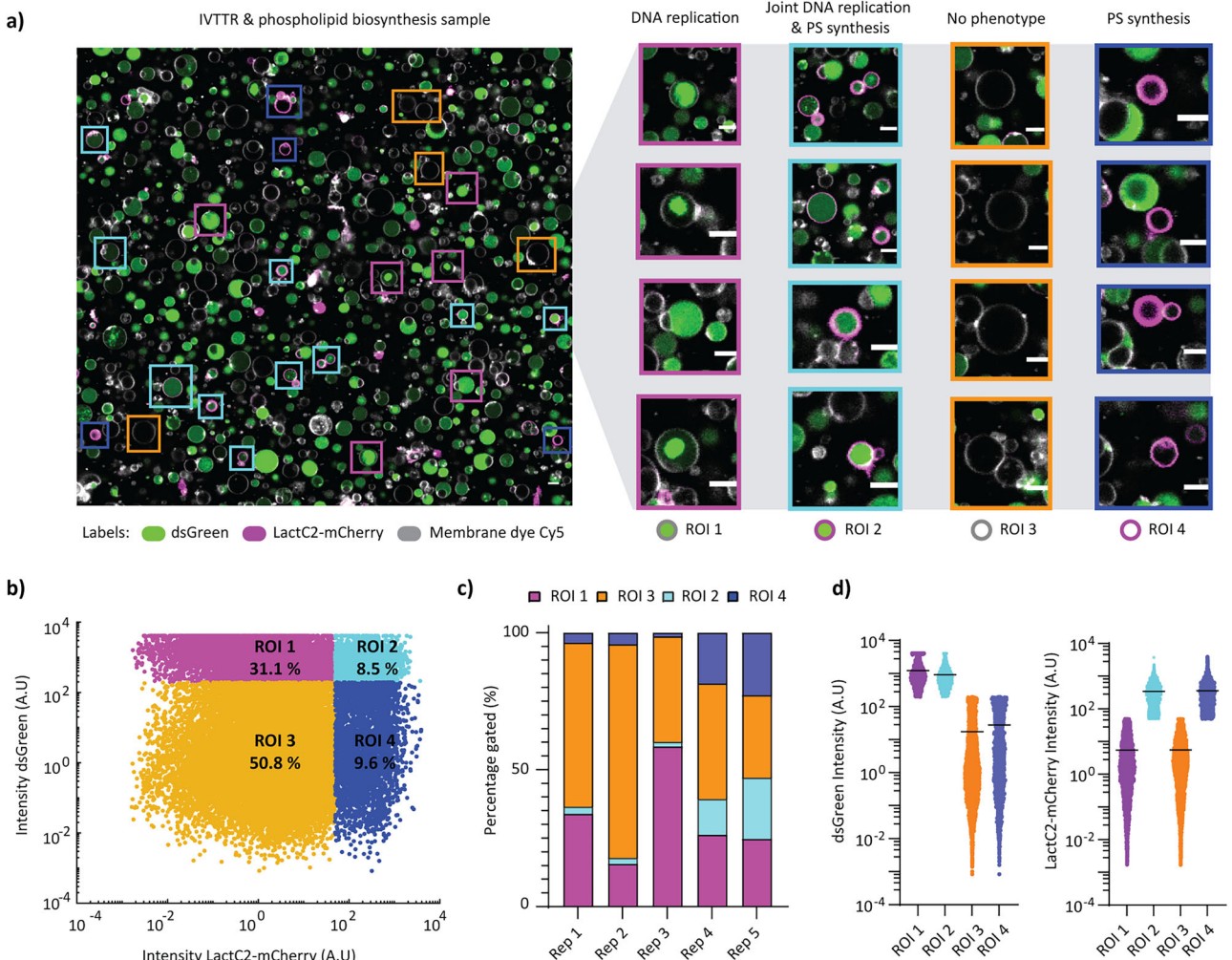

**Fig. 3 | High-content imaging of DNArep and PLsyn active phenotypes.**
**a** Confocal microscopy images of gene-expressing liposomes with complete DNArep and PLsyn reaction conditions. Membrane dye (Cy5) is colored in white, LactC2-mCherry in magenta, and dsGreen in green. Scale bar is 5 μm. Four distinct liposome phenotypes used for classification are highlighted: DNArep (ROI 1), dual DNArep and PLsyn (ROI 2), no module activity detected (ROI 3), and PLsyn (ROI 4). The cropped images correspond to the framed liposomes in the large panel.
**b** SMELDit image analysis on all biological repeats (~34,000 liposomes) builds a LactC2-mCherry vs. dsGreen phenotype map based on fluorescence intensity. Population subsets are gated into ROI 1-4 depending on the probe intensity, and are colored as in panel a. Percentages of liposomes per ROI are appended and were calculated from the pooled dataset. Phenotype maps from individual biological

repeats, as well as minus DNA negative control samples can be found in Supplementary Fig. 11. **c** Phenotype map (gated ROIs) from individual biological repeats. Dual phenotype region (ROI 2) is present in all replicates with at least ~100 identified liposomes. Specifically, 273 liposomes on Rep 1, 159 liposomes for Rep 2, 94 liposomes for Rep 3, 234 liposomes for Rep 4, and 2204 liposomes on Rep 5.
**d** Fluorescence intensity profiles from individual liposomes across all ROIs (panel b) suggest that DNArep activity remains unaffected when coupled with PLsyn activity (left graph), and vice-versa (right graph). Each dot represents a SMELDit-identified liposome. The number of liposomes is 10,851 (ROI 1), 2964 (ROI 2), 17,707 (ROI 3), and 3336 (ROI 4). Horizontal line indicates the mean of each data cluster. Source data are available for this figure in the Source Data file.

sulfate). We reasoned that possible inhibitory effects may arise by the substrates themselves, intermediate reaction products (e.g., lysophosphatidic acid, DOPA), or byproducts (e.g., Coenzyme A, deoxynucleoside monophosphate). Moreover, we hypothesized that DNA processing by the Φ29 DNA polymerase may either have a beneficial effect on PS synthesis by increasing the yield of synthesized enzymes through genome amplification[19] or have an adverse effect by hindering gene expression through collision events between DNA-interacting proteins (DSB or DNA polymerase vs. RNA polymerase)[21].

Following the same protocol as described above, we verified that DNArep and PLsyn were only active when their corresponding substrates/cofactors were present (Fig. 4b, c, and Supplementary Fig. 14), confirming that nonspecific staining with dsGreen and LactC2-mCherry was negligible. Using data pooled from all biological replicates, we found that the occurrence of DNArep-active liposomes (ROI 1 + 2) decreased only from ~38% to

~31% when PLsyn substrates were supplied, while the occurrence of PLsyn-active liposomes (ROI 2 + 4) reduced only from ~18% to ~15%/~10% (with/out dNTPs) when DNArep substrates/cofactors were supplemented (Fig. 4d and Supplementary Fig. 15). By examining individual replicates, we found a higher variability on the occurrence of PLsyn-active liposomes when reactions contained all DNA replication substrates/cofactors (both modules ON) compared to in their absence (Supplementary Fig. 15), but its cause remains to be explained. Moreover, the intensity distributions of DNArep and PLsyn activity reporters were similar regardless of the presence or absence of the substrates from the other module (Fig. 4e). Furthermore, DNA replication efficiency was similar with or without the substrates for PLsyn (Fig. 2f and Supplementary Fig. 16). Overall, we conclude that functional integration of the DNArep and PLsyn modules is minimally affected by metabolic crosstalk or by module co-activity.

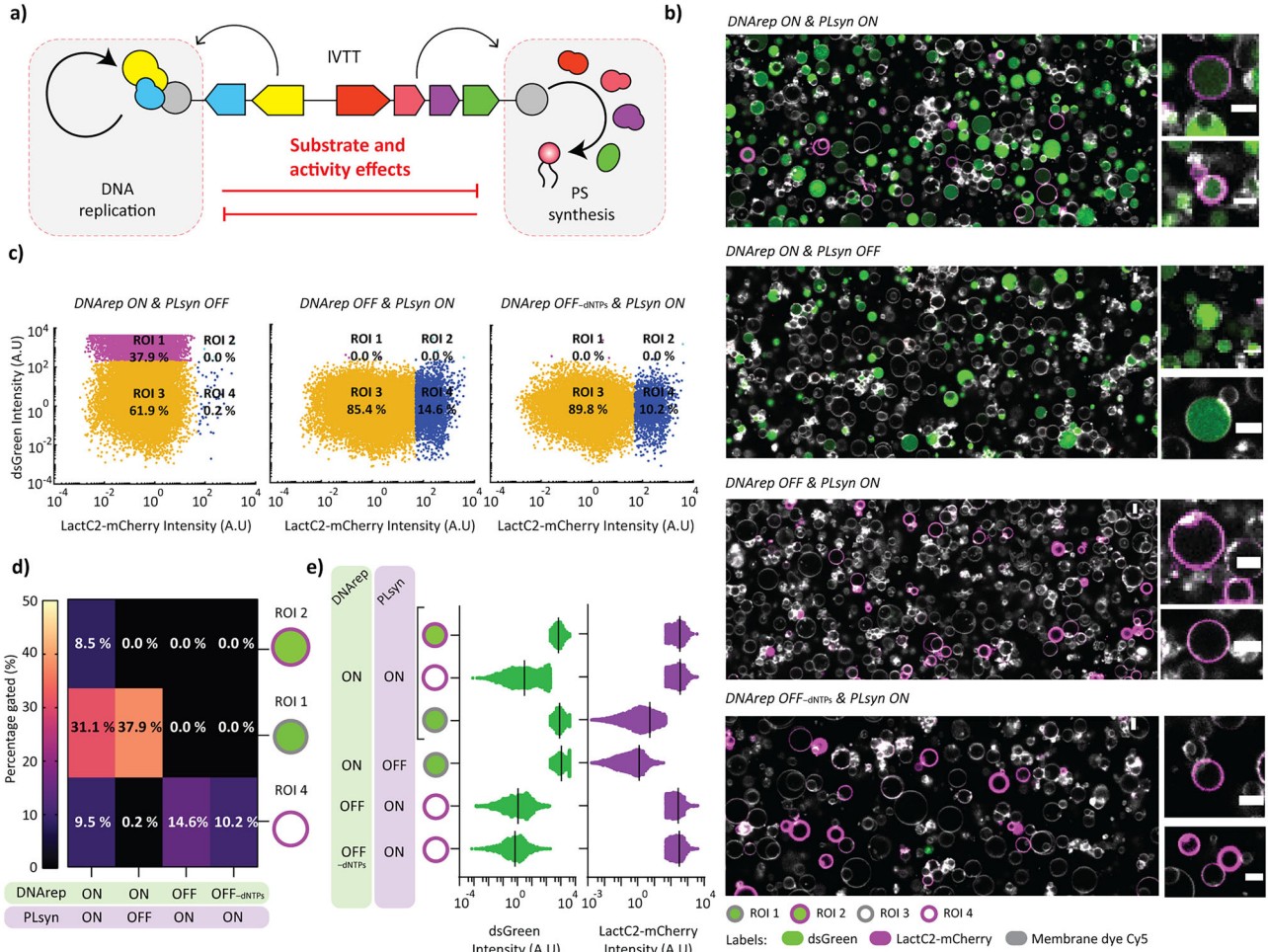

**Fig. 4 | Effects of turning a module ON and OFF on the activity of the other module. a** Schematic of co-active DNArep and PLsyn with an emphasis on metabolic and activity crosstalk effects. **b** Confocal microscopy images of liposome samples show that different substrate additions trigger a specific module activity. ON and OFF labeling indicates presence (ON) or absence (OFF) of substrates/cofactors for either DNArep or PLsyn. Liposome membrane dye (Cy5) is colored in white, LactC2-mCherry in magenta, and dsGreen in green. Scale bar is 5 μm. **c** Phenotype scatter plots from SMELDit image analysis (LactC2-mCherry vs. dsGreen) on all biological repeats (*n* = 3) show only one active module if substrates are omitted for the other one (ON or OFF state). Classified liposome subpopulations are labeled as ROI 1-4 and gated in different colors as in Fig. 3b. Appended percentages are calculated from the pooled dataset including all biological repeats.

Scatter plots from the individual repeats can be found in Supplementary Fig. 12. **d** Phenotype heatmap with gated percentage values for ROIs 1, 2, and 4 calculated across all replicates with active and/or inactive modules indicates minor crosstalk between the activity of the DNArep and PLsyn modules. Percentages for individual repeats can be found in Supplementary Fig. 12. **e** dsGreen and LactC2-mCherry intensity profiles across gated ROIs are similar under both single or joint-module activity. Each dot represents a SMELDit identified liposome. The number of liposomes is 10,851 (ROI 1, full reaction), 7934 (ROI 1, DNArep), 2964 (ROI 2, full reaction), 1686 (ROI 4, PLsyn –dNTPs), 2606 (ROI 4, PLsyn), 3336 (ROI 4, full reaction). Vertical lines indicate the mean value of each data cluster. Source data are available for this figure in the Source Data file.

## Influence of the genetic context on module activity

Next, we investigated whether the genetic background could influence the activity of a module. For this, we compared liposome populations with *DNArep-PLsyn* genome against liposomes with DNA templates carrying only the genes of a single module, i.e., either *DNArep* or *PLsyn*, in the presence of the full set of substrates and cofactors (Fig. 5a). We hypothesized that module activity from the *DNArep-PLsyn* genome may be compromised by sharing of resources/machinery allocated to gene expression[23], by impaired replication caused by strand switching of polymerizing DNAP, or by collision events between DNA interacting proteins (DNA and RNA polymerases)[21]. All three effects would become more prominent as the number of genes increases. Alternatively, genome amplification may boost lipid biosynthesis by increasing the concentration of PLsyn enzymes[19].

As expected, microscopy images showed that the appearance of liposome phenotypes was directed by the encapsulated DNA program (Fig. 5b,c, and Supplementary Fig. 17). Interestingly, the percentages of

DNArep-positive liposomes were similar with and without co-expression of the *PLsyn* genes, decreasing only from ~45% (ROI 1) to ~40% (ROI 1 + 2) when *PLsyn* was co-expressed (Fig. 5d). Conversely, the percentages of PLsyn-positive liposomes dropped from ~38% (ROI 4) to ~18% (ROI 4 + 2) when *DNArep* was co-expressed (Fig. 5d and Supplementary Fig. 18), suggesting that PLsyn activity is more sensitive to genetic background and expression burden than DNArep activity. This effect may also limit dual-module activity in liposomes containing *DNArep-PLsyn*, explaining the higher prevalence of a single phenotype in PLsyn- and DNArep-containing liposomes (ROI 4, ~38% on PLsyn and ~44% on DNArep), compared to those with DNArep-PLsyn displaying both phenotypes (ROI 2, ~8%) (Fig. 5d).

While the occurrence of liposomes exhibiting an active PLsyn module was influenced by the co-expression of *DNArep*, we noticed that the intensity distributions reporting the levels of DNArep and PLsyn activity were similar under single- and double-genetic module expression conditions (Fig. 5e). For PLsyn, this suggests that, above a

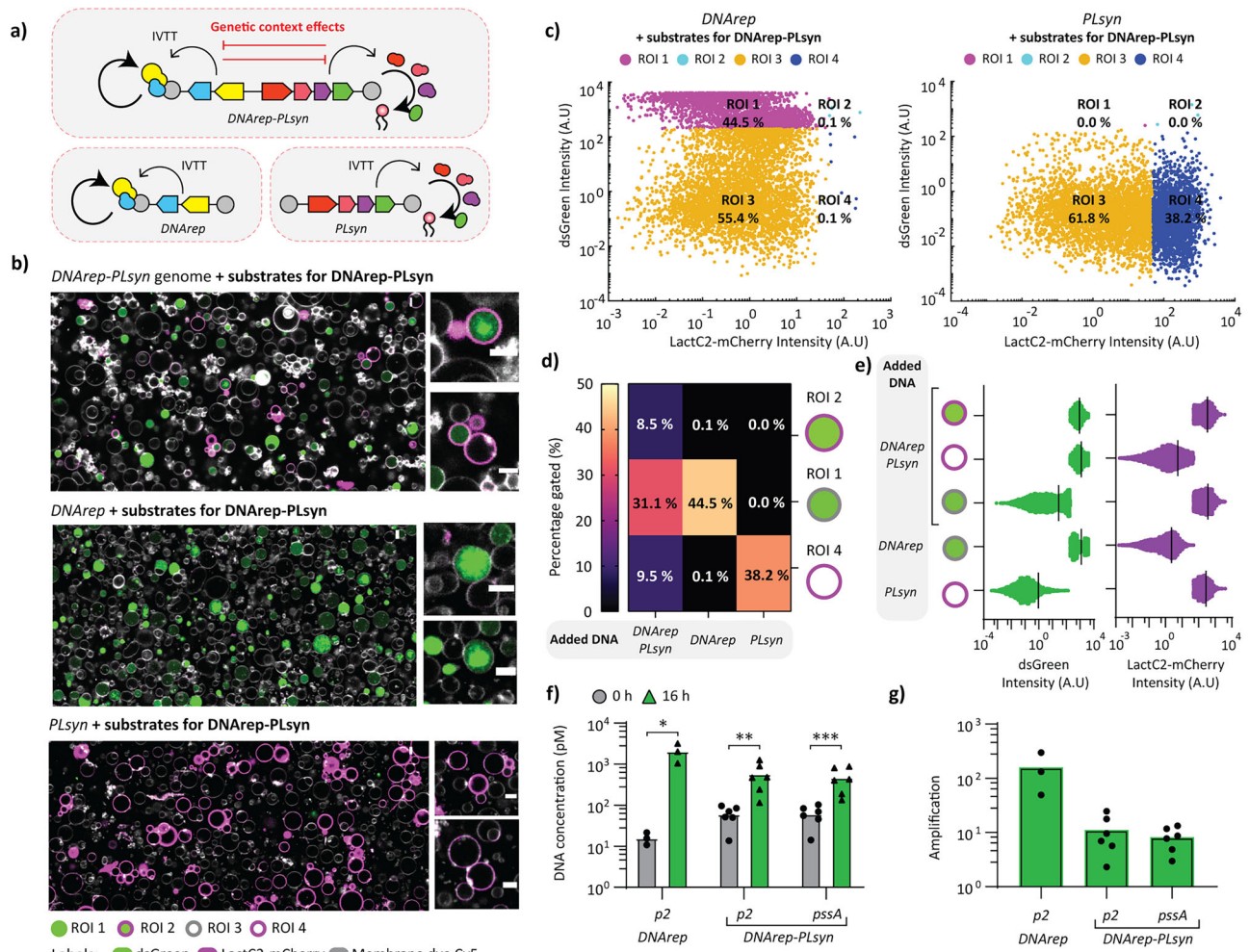

**Fig. 5 | Effects of DNA template and co-expression of genetic modules on DNArep and PLsyn activity. a** Schematic of the expression of the coupled (top) and separate (bottom) genetic modules from specific DNA templates. The comparison leverages the influence of genetic context on module activity. **b** Confocal microscopy images of liposome samples with all substrates and cofactors show DNA-specific phenotypic outputs. The used templates are indicated. Liposome membrane dye (Cy5) is colored in white, LactC2-mCherry in magenta, and dsGreen in green. Scale bar is 5 μm. **c** Phenotype scatter plots from SMELDit image analysis (LactC2-mCherry vs. dsGreen) on the biological repeats show only one phenotype output for each DNA program, DNArep or PLsyn. Classified liposomes in ROI 1-4 are gated in different colors. Appended percentages are calculated from the pooled data of biological repeats ($n = 1$ for DNArep and 2 for PLsyn). Scatter plots from individual biological replicates can be found in Supplementary Fig. 17. **d** Phenotype heatmap constructed from all repeats with different template conditions: *DNArep-PLsyn*, *DNArep* and *PLsyn* DNAs. **e** dsGreen and LactC2-mCherry intensity profiles across all ROIs have similar distributions. Each dot represents a SMELDit identified liposome. The number of liposomes is 3336 (ROI 1, *DNArep-PLsyn*), 3400 (ROI 1,

*DNArep*), 2964 (ROI 2, *DNArep-PLsyn*), 2,294 (ROI 4, *PLsyn*), 10,851 (ROI 4, *DNArep-PLsyn*). Vertical lines indicate the mean of each data cluster. **f** Absolute DNA quantification from liposome samples show higher DNA replication yields for the minimal self-replicator *DNArep* when compared with the *DNArep-PLsyn* genome. The targeted regions on the *pssA* and *p2* genes are indicated. DNA concentration changes between 0 h and 16 h were assessed using a two-sided log-ratio paired *t*-test. Log-transformed ratios of 16-h to 0-h values were calculated for each replicate and each corresponding gene (*pssA* and *p2*). A one-sample *t*-test was then performed to determine if the mean log-ratio significantly differed from zero ($p < 0.05$). * $p \leq 0.05$, ** $p \leq 0.01$, and *** $p \leq 0.001$. Exact *p* values are 0.0114 (*p2* in *DNArep*), 0.00162 (*p2* in *DNArep-PLsyn*), and 0.00035 (*pssA* in *DNArep-PLsyn*). **g** Amplification fold of *DNArep-PLsyn* and *DNArep* DNA templates calculated from qPCR data in panel f: end-point (16 h) DNA concentration / DNA concentration at time zero. In f and g data points represent biological repeats ($n = 3$ or 6) and bar height the mean value. Source data are available for this figure in the Source Data file.

detectable activity threshold, DOPS production yield was not affected by co-expression of *DNArep* genes. For DNA replication, however, dsGreen signal intensity is proportional to DNA quantity, which accounts for DNA length and amplification fold. Therefore, similar dsGreen intensities from replicated *DNArep-PLsyn* and *DNArep* templates suggest that the amplification fold of *DNArep-PLsyn* is lower than that of *DNArep* given its larger size (~9.6 kb vs. ~3.2 kb). To test this hypothesis, we performed absolute DNA quantitation by qPCR, confirming that *DNArep-PLsyn* replicates at a lower yield than *DNArep* (~10-fold vs. ~100-fold) (Fig. 5f,g). Considering that the yield of synthesized DNAP and TP does not differ much from the *DNArep-PLsyn* or *DNArep* templates (in the absence of module-specific substrates and cofactors)

(Fig. 1e and Supplementary Fig. 3), we speculate that the processivity of or polymerization by DNAP—and not replication initiation—might be the amplification bottleneck, especially under transcribing conditions. This hypothesis is supported by previous observations that up to a length of 6 kb the rate limiting step is initiation; over 6 kb, the DNA length becomes rate limiting[24].

We next examined how phenotype appearance developed in the course of gene expression. Flow cytometry data show that the highest percentage of liposomes with joint phenotypes was reached at a later time compared to liposomes containing the genes of a single module (8 vs. 4 h) (Supplementary Fig. 19-21). Time course analysis of DNA replication by qPCR showed that the maximum amplification fold was

reached after 4 h for both the *DNArep-PLsyn* and *DNArep* templates (Supplementary Fig. 21). This mostly reflects the DNA replication kinetics in the larger population of PLsyn-inactive liposomes expressing the full genome (ROI 1). These results demonstrate that some genetic factors slow down the dynamics of template replication when the PLsyn module is concurrently active.

## Setting up the stage for integrative evolution

Finally, we envisioned that module performance and integration could be enhanced through directed evolution. Evolving *DNArep-PLsyn* for increased functionality, e.g., a higher yield of synthesized phospholipids or faster appearance of the combined modules, would require a recursive cycle of genome library encapsulation and expression−phenotype interrogation−sorting of liposomes with the desired features−DNA recovery and amplification. We hereby streamlined the key experimental steps that are required for laboratory evolution (Supplementary Fig. 22).

First, to facilitate handling of DNA across the different stages, we utilized a yeast-assembled plasmid (pY003 or pY005) as a precursor of the linear *DNArep-PLsyn* template. A cloned and sequence-verified plasmid provides a more stable template for the fast production of DNA libraries. High amounts of the PCR-amplified genome could be obtained, allowing for nanomolar input concentrations. Its expression in liposomes improved the robustness and efficiency of the system, yielding ~30% (vs 8%) of fully active synthetic cells and over 37% (vs 17%) with at least one functional module, while reducing variability between replicates (Supplementary Figs. 23–25). The change of phenotype map compared to Fig. 3c (template obtained by overlapping PCR) suggests an increased ability of DNArep-active liposomes to also produce phospholipids. Time-lapse imaging of phenotype development shows that DNArep precedes PLsyn and the activity of both modules reaches a plateau after 6 h (Supplementary Fig. 26).

Second, to establish a tight coupling between genotype and phenotype, we reduced the concentration of *DNArep-PLsyn* genome down to 50 pM, which corresponds to an expected average copy number of DNA per liposome equals to one[19]. Under these conditions a significant fraction of liposomes exhibiting combined module activation was still detected (Supplementary Fig. 22). Next, we screened liposomes and sorted those identified in ROI 2 by fluorescence activated cell-sorting (FACS) (~3000 events). The full-length genome was successfully recovered and amplified by PCR (Supplementary Fig. 22), and could serve as a template to start a new round. These data validate that an entire cycle of evolutionary engineering is feasible for improvement of integrated biological functions within a DNA-driven synthetic cell model.

## Discussion

This work shows how genetically encoded functions can be integrated in a synthetic cell model. Self-replication of a DNA genome and enzymatic phospholipid synthesis were driven by a minimal transcription-translation system emulating the logic of cellular life. Although we routinely obtained more than a hundred vesicles with coupled module activities per sample, these only represent about 8% of the total liposome population when using a DNA template generated by overlapping PCR. We suspected that the robustness of the system may be limited by the DNA quality and purity. Using a stable source of DNA, namely a sequence-verified circular plasmid to generate the linear *DNArep-PLsyn* template, improved the system's efficiency over a factor of 3.5 with ~30% of fully functional synthetic cells. We attribute this improvement to a higher purity of the full-length DNA that also enables to work with increased concentrations of active template. The remaining fraction of nonfunctional synthetic cells probably arises from deficient molecular compositions inside vesicles, which cannot support all functions concurrently. For instance, differences in the loading of DNA or translation machinery between liposomes probably lead to varying levels of expressed genes, while uneven supply of substrates or cofactors for DNArep and PLsyn would result in varying outputs of the modules. Even if it is clear that prototype synthetic cells will not have the level of control and robustness of natural living cells, a major challenge will be to mitigate this heterogeneity and increase the proportion of active vesicles.

Besides, challenges remain to achieve physical growth of liposomes upon lipid biosynthesis. While the DNA amplification fold may be sufficient for the partitioning of the genetic material amongst daughter vesicles upon division, the bottleneck for a cell cycle is on the yield of synthesized phospholipids. With the produced DOPS representing only <7% of the total lipid amount, the current system cannot support physical expansion of the membrane. Using purified, detergent-solubilized enzymes reconstituted into large vesicles, Exterkate et al. showed efficient conversion of the fatty acid precursor, which enabled membrane growth[25]. In other studies, de novo membrane synthesis was demonstrated with phospholipid analogs produced by native chemical ligation[3,26]. The compatibility of these approaches with PURE system driving the expression of an internal DNA program remains to be investigated.

We found that the DNA replication and DOPS synthesis processes are fully compatible; they can simultaneously be operated without interfering with each other. However, insertion of a second genetic module reduced the occurrence of liposomes with DOPS production or the yield of amplified DNA compared to the situation in which a single genetic module was present. To alleviate this genetic burden, different designs of the *DNArep-PLsyn* genome could be tested to optimize the metabolic balance and resource allocation for gene expression. For instance, gene organization in the form of operons[27], or a different combination of regulatory elements[27–30] could be attempted. Considering that translation is a gene-expression bottleneck[31], ribosome binding sites (RBSs) of different strengths could be scanned for achieving balanced expression of the DNArep and PLsyn machineries[32,33]. Stringent temporal control over gene expression may also be realized by implementing genetic circuits, for example ON/OFF switches regulated by specific signals[34,35]. With this, the processes of genome replication and membrane synthesis could be separated in time, reducing competition for resources and possible clashes between DNA processing enzymes. Lastly, protein properties could be ameliorated through engineering by mutagenizing the DNA coding sequence. For example, encoding a Φ29 DNAP with higher processivity may increase the replication yield of long genomes[36].

While some of these modifications can be realized by rational design, we also propose to use directed evolution as an engineering tool to enhance synergy of the DNArep and PLsyn modules[11]. We postulate that integration of rudimentary functions, such as DNArep and PLsyn, followed by evolutionary engineering, is a more effective *modus operandi* than optimizing the individual modules separately prior to combining them. Genetic diversification of the *DNArep-PLsyn* genome may occur through multiple rounds of template replication enabling the fixation of advantageous mutations directly inside liposomes[15,37]. Alternatively, random or targeted mutations could be externally introduced, e.g., by PCR or recombineering methods[38], and the library of genome variants could be used as the DNA input to start an evolution cycle (Supplementary Fig. 22).

Other interesting extensions of our work include the interconnection between the different subsystems[39], and the integration of more cellular modules[4,5,8,40] followed by system's level evolution[11]. Finally, incorporating more advanced computational methods, like unsupervised learning[41], for segmentation, feature extraction, and phenotype classification would enhance reproducibility of the analysis, while expanding the discovery of properties.

## Methods

### Buffers and chemicals

All buffers were prepared with MilliQ grade water with 18.2 MΩ resistivity (Millipore, USA). All chemicals were purchased from Sigma-Aldrich unless indicated otherwise.

### DNA construct design

Plasmids G363 and G435 utilized for *DNArep-PLsyn* genome assembly were derived from previously cloned constructs encoding for PL synthesis enzymes (pGEMM7)[2] or for Φ29 DNA replication proteins (G95)[7]. Plasmid G555 harboring the Φ29 ori-flanked *PLsyn* gene pathway was constructed by subcloning the four-gene *PLsyn* fragment from G363 (digested with Xho1 and Nco1 restriction enzymes), into a Φ29 origins flanked vector[7], also digested with Xho1 and Nco1. PCR fragments for *DNArep-PLsyn* genome assembly were prepared from G363 with 5'-phosphorylated 491 ChD and 1302 ChD primers (*PLsyn_{frag}*), and from G435 with 5'-phosphorylated 492 ChD and 1289 ChD primers (*DNArep_{frag}*). Linear DNA fragments containing either of the two genetic modules were prepared by PCR using 5'-phosphorylated primers 491 and 492 ChD, and KOD Xtreme Hot Start DNA polymerase. The reaction solution contained (final concentrations) 1× Extreme Buffer (MERCK), 0.02 U/μL KOD DNA polymerase (MERCK), ~0.3 ng/μL DNA template, 300 nM forward and reverse primers, 0.4 mM of each dNTP, and MilliQ water up to 50 μL final volume. The thermocycler protocol was set to 94 °C for 2 min for polymerase activation, followed by 25–30 cycles of 10 s 98 °C, 20 s 60 to 68 °C, 30–60 s/kb at 68 °C depending on the length of the desired amplicon.

When specified, a negative control genome was used. In addition to the *PLsyn* module used for the construction of the *DNArep-PLsyn* genome, plasmid G363 encodes the *p2* and *p3* genes from Φ29 under the control of SP6 promoters. Plasmid G363 was linearized with primers 1119 ChD and 756 ChD to generate a 9627-bp amplicon capable of expressing PlsC, CdsA, PssA, and a 140-amino acid long truncated PlsB. Without the addition of SP6 RNA polymerase, both modules are inactive. The PCR reaction mix consisted of 1× KOD One Master Mix (TOYOBO), 300 nM of each forward and reverse primers (491 and 492 ChD), and ~0.1 ng μL⁻¹ of plasmid DNA. The thermocycler protocol consisted of 35 cycles of 10 s at 98 °C and 50 s at 68 °C.

All PCR amplicons were verified for correct DNA length by 0.7-1% agarose gel electrophoresis before PCR clean-up with QIAquick PCR purification kit (Qiagen). Whenever needed, DNA was purified directly from the agarose gel with a QIAquick Gel Extraction Kit (Qiagen). Both PCR and gel purification standard protocols were modified with a longer column drying step (~5 min at 10,000 g) before DNA elution with MilliQ water. Purified DNAs were quantified by Nanodrop 2000c spectrophotometer (Isogen Life Science).

### In vitro assembly of the *DNArep-PLsyn* genome

*DNArep-PLsyn* genome was constructed by overlap PCR from *DNArep* and *PLsyn* fragments with two main PCR steps: (i) plasmid overlap and full product DNA extension, and (ii) addition of 491 and 492 ChD primers and PCR amplification of the full *DNArep-PLsyn* product. 47 μL reactions were prepared with final concentrations of 1× Extreme Buffer (MERCK), 0.02 U/μL KOD DNA polymerase (MERCK), ~0.3 ng μL⁻¹ DNA template, 0.4 mM of each dNTP, and MilliQ water. The thermocycler was programmed for 2 min at 94 °C for polymerase activation, followed by five cycles of (10 s at 98 °C, 20 s at 60 °C and 3 min and 40 s at 68 °C), and 20 cycles of (10 s at 98 °C, 7 min at 68 °C). For the last 20 cycles of the overlap PCR program, primers 491 and 492 ChD were added to the reaction to a final concentration of 300 nM each. The list of primers is reported in Supplementary Table 2.

### In vivo assembly of *DNArep-PLsyn* in yeast

In vivo assembly in *S. cerevisiae* was used to build plasmids pY003 and pY005 containing the *DNArep-PLsyn* genome. pY003 was constructed from three DNA fragments: *DNArep*, *PLsyn*, and a fragment containing the yeast centromeric origin of replication *CEN6/ARS4* and the auxotrophic marker *URA3* for maintenance and selection in yeast. The *PLsyn* fragment was amplified from G363 with primers 41 ChDT and 58 ChDT, the *DNArep* fragment from G435 with primers 1289 ChD and 44 ChDT, and the *CEN6/ARS4-URA3* fragment from pRS316 with primers 56 ChDT and 57 ChDT. For pY005, the *CEN6/ARS4-URA3* fragment was replaced by a fragment that also included an *E. coli* origin of replication and an ampicillin resistance marker. The *pUC-Amp-CEN6/ARS4-URA3* fragment was PCR-amplified from pRS316 using primers 57 ChDT and 64 ChDT that were designed to introduce 60-bp overlapping ends for efficient in vivo assembly. The PCR reaction mix contained (final concentrations) 1× Extreme Buffer (MERCK), 0.02 U μL⁻¹ KOD DNA polymerase (MERCK), ~0.2 ng μL⁻¹ DNA template, 200 nM forward and reverse primers, 0.4 mM of each dNTP, and MilliQ water up to 50 μL final volume. The thermocycler protocol was set to 94 °C for 2 min for polymerase activation, followed by 35 cycles of 15 s at 98 °C, 20 s at 60 °C, and 60 s per kb at 68 °C depending on the length of the desired amplicon. PCR amplicons were analyzed by agarose gel electrophoresis and purified as described above. The DNA fragments were then pooled at a concentration of 100 fmol for the *CEN6/ARS4-URA3* fragment and 200 fmol for the other two DNA fragments. Transformation in *S. cerevisiae* CEN.PK2-1C was carried out using the lithium acetate/single-stranded carrier DNA/polyethylene glycol method[42]. For selective growth, cells were plated on complete supplemental medium without uracil (CSM-URA) agar plates composed of 6.7 g L⁻¹ yeast nitrogen base without amino acids (Formedium), 20 g L⁻¹ glucose (Formedium), 0.77 g L⁻¹ CSM-URA (MP Biomedicals), and 20 g L⁻¹ agar (Euromedex). Plates were incubated at 28 °C for 2 days.

For pY003 and pY005 isolation from *S. cerevisiae*, 15 mL cultures were grown in liquid CSM-URA media at 28 °C with shaking at 200 rpm (Infors AG CH-4103 Bottmingen). Cells were harvested at mid-log phase by centrifugation at 3800 g for 5 min at 4 °C. Cell pellets were then washed with 20 mL of 100 mM Tris, pH 8.0, and centrifuged again at 3800 g for 5 min at 4 °C. The pellets were resuspended in 60 μL of a freshly prepared solution containing 100 mM Tris (pH 8.0), and 10 mM DTT. To this, 3 μL of RNase A (10 mg mL⁻¹) were added, and the suspension was incubated at 30 °C for 10 min with shaking at 300 rpm (Eppendorf thermomixer comfort). After incubation, 8 μL of a solution containing 100 mM Tris (pH 8.0), 10 mM DTT, and 2 U μL⁻¹ Zymolyase (20 T, from *Arthrobacter luteus*, Amsbio) were added to the cell suspension. The mixture was incubated again at 30 °C for 30 min with shaking at 300 rpm. Spheroplasting efficiency was evaluated by measuring the optical density at 660 nm (Cary 100 Scan UV-Visible Spectrophotometer) of the samples before and after Zymolyase treatment, with an 70–100% reduction in OD indicating successful spheroplast formation. Finally, plasmid DNA was extracted using the EZ-10 Spin Column Plasmid DNA Miniprep Kit, following the manufacturer's instructions starting from the addition of Solution II (lysis buffer). DNA was eluted in 25 μL of prewarmed MilliQ water.

Plasmid pY005 was re-transformed in *E. coli* DH5α chemically competent cells. Transformed cells were recovered in 1 mL LB medium for 1 h and were then plated on 50 μg mL⁻¹ ampicillin-LB agar plates. Following overnight incubation at 37 °C, a colony was picked to inoculate 5 mL 100 μg mL⁻¹ ampicillin - LB medium. The liquid culture was grown at 37 °C and 280 rpm overnight. Plasmid was purified from 3 mL of liquid culture using the PureYield Plasmid MiniPrep System (Promega). DNA quality and concentration were determined via Nanodrop 2000c spectrophotometer (Isogen Life Science) followed by sequence verification by Oxford Nanopore sequencing (Plasmidsaurus). Three single-nucleotide substitutions were identified, two silent mutations in the *p2* gene at positions E11 and K179, and mutation D28Y in the *pssA* gene. While no reports on inhibitory effects due to this mutation exist in literature, our results demonstrate that it does not abolish phosphatidylserine synthase activity.

Linear DNA fragments compatible with phi29-based DNA replication were obtained by nested PCR from pY003. The first PCR reaction (outer PCR) was performed using primers that annealed externally to the *oriL* and *oriR* flanks (50 ChDT and 51 ChDT). Subsequently, 1 μL of the product of the first PCR was used as the template for the second PCR reaction (inner PCR) using the 5'-phosphorylated primers 491E ChDT and 492E ChDT. The PCR reaction mix consisted of (final concentrations) 1× Extreme Buffer (MERCK), 0.02 U/μL KOD DNA polymerase (MERCK), 1 μL of template (Outer PCR: 1 μL of yeast-isolated plasmid; Inner PCR: 1 μL of the outer PCR), 300 nM forward and reverse primers (Outer PCR, 50 ChDT and 51 ChDT; Inner PCR, 491E ChDT and 492E ChDT), 0.4 mM of each dNTP, and MilliQ water up to 25 μL final volume. The thermocycler protocol was set to 94 °C for 2 min, followed by cycles of 10 s at 98 °C, 20 s at 60 °C, and 7.5 min at 68 °C (Outer PCR, 20 cycles; Inner PCR, 35 cycles).

For activity assays reported in Supplementary Fig. 23 a KOD polymerase variant with reduced mutagenesis rate, KOD One (TOYOBO), was used to generate linear templates from pY005. The PCR reaction mix consisted of 1× KOD One Master Mix, 300 nM of each forward and reverse primers (491 and 492 ChD), and ~0.1 ng μL$^{-1}$ of *E. coli* isolated plasmid DNA. The thermocycler protocol consisted of 35 cycles of 98 °C for 10 s and 68 °C for 50 s. All PCR amplicons were analyzed by agarose gel electrophoresis and purified as described earlier

The lists of plasmids and primers are reported in Supplementary Tables 1 and 2, respectively.

## Purification of SSB, DSB, and LactC2-mCherry

Φ29 SSB and DSB were expressed and purified as described in Ref. 43. and Ref. 17, respectively. Briefly, SSB was purified from *B. subtilis* cells infected with Φ29 bacteriophage. The cells were lysed, the proteins were extracted with buffer A (50 mM Tris-HCl (pH 7.5), 5% glycerol, 1 mM EDTA, 7 mM β-mercaptoethanol (BME)) supplemented with 0.2 M NaCl and precipitated from the supernatant with 65% ammonium sulfate. The pellet was dissolved in buffer A and the SSB-containing pellet was resuspended in a mixture of buffer A supplemented with 50% glycerol and 65% ammonium sulfate. The sample was centrifuged, the pellet discarded, and the supernatant diluted up to a conductivity of 4.2 KΩ$^{-1}$ cm$^{-1}$ and loaded onto a coupled phospho-cellulose-DEAE-cellulose column. Following dialysis, the SSB stock concentration was 10 mg mL$^{-1}$, stored in a buffer with 50 mM Tris, pH 7.5, 60 mM ammonium sulfate, 1 mM EDTA, 7 mM BME, and 50% glycerol.

DSB was heterologously expressed in *E. coli* BL21 (DE3) cells. The cell pellet was lysed and resuspended in buffer A supplemented with 0.8 M NaCl. After centrifugation and removal of DNA with polyethyleneimine, the supernatant was diluted with an equal volume of buffer A supplemented with 0.4 M NaCl and 0.04% polyethyleneimine. Following centrifugation, the DSB-containing pellet was resuspended with buffer A plus 1 M NaCl by stirring vigorously. The protein was precipitated with ammonium sulfate at 70% saturation, the pellet was resuspended in buffer A supplemented with 25% glycerol and loaded onto a phosphocellulose Whatman P11 column. The eluted protein sample was loaded on a Q Sepharose column and the eluted DSB was precipitated with ammonium sulfate at 70% saturation. After centrifugation, the pellet was resuspended to reach a DSB stock concentration of 10 mg mL$^{-1}$, stored in a buffer with 50 mM Tris, pH 7.5, 0.1 M ammonium sulfate, 1 mM EDTA, 7 mM BME, and 50% glycerol.

LactC2-mCherry protein was expressed and produced as described in Ref. 19. Briefly, histidine-tagged LactC2-mCherry was expressed in *E. coli* BL21(DE3) (NEB). The cell pellet was resuspended in lysis buffer, disrupted by sonication on ice, and centrifuged to remove the cell debris. The proteins were purified with HisPure Ni-NTA resin (Thermo Scientific). After elution, the fluorescent fractions were buffer-exchanged with storage buffer (50 mM HEPES-KOH, pH 7.5,

150 mM NaCl, 10% glycerol) using a 10-MWCO Amicon Ultra-15 centrifugal filter unit (Merck). The concentration was determined with a Bradford assay and the protein was stored in −80 °C. Before liposome staining, the protein stock vial was centrifuged at maximum speed (Eppendorf Centrifuge 5415 R) for 10 min to spin down protein aggregates. If necessary, LactC2-mCherry protein stock was diluted in homemade PURE buffer (PB) consisting of 180 mM potassium glutamate monohydrate, 14 mM magnesium acetate tetrahydrate, and 20 mM HEPES at a pH of 7.6 (adjusted with potassium hydroxide) before usage.

## In-liposome gene expression

Lipid-coated beads were prepared as explained in Refs. 2,7 with minor modifications. A lipid mixture was prepared with chloroform-dissolved lipids (Avanti Polar Lipids) in a 5-mL round-bottom flask. The solution contained 49.5% DOPC, 33.7% DOPE, 12% DOPG, 3.8% 18:1 CL, and 1% DSPE-PEG-biotin mass composition, for a total mass of lipids of 2.02 mg. For confocal microscopy experiments the primary lipid composition was adjusted to include 0.05% mass of DOPE-Cy5. The resulting mix was supplemented with 25.4 μmol of rhamnose (Sigma-Aldrich) dissolved in methanol. 600 mg of 212–300 μm glass beads (acid-washed) were added to the 5 mL round-bottom flask containing the lipid/rhamnose solution, and the solvent was evaporated with a rotary evaporator (Heidolph) for 2 h at room temperature and 200 mbar. The lipid-coated beads were recovered and further dried in a desiccator overnight. The dried lipid-coated beads were flushed with argon and stored at −20 °C until use. In 1.5 mL Eppendorf tubes, PURE*frex2.0* (GeneFrontier) reactions were assembled as recommended by the manufacturer using 500 pM DNA template (if not mentioned otherwise) and 0.75 U μL$^{-1}$ of Superase·In RNase inhibitor (Thermo Fisher). DNA assembled in vitro was used in all experiments, except for those described in Supplementary Figs. 4,22–26, where the genome assembled in yeast was employed. When indicated, the IVTT mixture was supplemented with the required substrates and cofactors for DNA replication (300 μM dNTPs, 20 mM ammonium sulfate, 0.75 mg mL$^{-1}$ SSB, and 0.21 mg mL$^{-1}$ DSB, final concentrations) or/and PL synthesis (1 mM CTP, 500 μM G3P, 500 μM L-Serine, and 5 mM BME, final concentrations, the oleoyl-CoA precursor was added in a next step described below). 10 mg of lipid-coated beads freshly desiccated for 20-30 min were added to 20 μL of swelling solution. The sample in Eppendorf tube was kept in a 4 °C room for gentle rotation with an automatic tube rotator (VWR) for 30 min and was then subjected to four freeze-thaw cycles with alternating steps of dipping in liquid nitrogen and thawing on ice. Liposomes were recovered with a cut pipette tip and transferred to a PCR tube with DNase One (NEB) added to a final concentration of 0.1 U μL$^{-1}$. For PLsyn activity, the sample was further transferred to another PCR tube with a pre-deposited O-CoA (Avanti Polar Lipids) film. The dried precursor film was prepared by dissolving the O-CoA powder in chloroform:methanol:water (40:10:1 vol. fractions), where after a solution containing 1 μg O-CoA was transferred to a PCR tube and the solvent was evaporated at ambient temperature and pressure. With the added liposome suspension, the final O-CoA concentration was 176 μM, and when needed (i.e., liposome suspension volumes changed), the pre-deposited O-CoA quantity was adjusted to maintain the same final concentration. Samples were incubated at 30 °C, 37 °C, or 34 °C for 16 h. A maximum of 15 μL liposome suspension was handled per PCR tube. If higher volumes were needed, the suspension was distributed across different PCR tubes with independently added O-CoA dried lipid film.

## Flow cytometry

In a PCR tube, a liposome suspension (1.5 μL) was mixed in a 1:1 ratio with a 1000× diluted dsGreen (Lumiprobe) stock solution in PB and the sample was left to incubate in the dark at room temperature for 30 min. The 3 μL liposome-dsGreen mixture was further diluted by

adding 197 μL of a 1:1000 dsGreen:PB solution. To remove possible remaining glass beads, the 200 μL solution was filtered through a 35 μm nylon mesh of a cell-strainer cap from 5 mL round-bottom polystyrene test tubes (Falcon). With a large volume pipette tip, 138.5 μL of the filtered solution was transferred to a 2 mL round-bottom tube to which 1.5 μL of 1:1000 dsGreen:PB solution and 10 μL of LactC2-mCherry probe were added to obtain a final LactC2-mCherry concentration of 300 nM and a final volume of 150 μL. The sample was incubated for 1 h before injection in a FACSCelesta flow cytometer (BD Biosciences) set up with a 488-nm laser and 530/30 filter for detection of dsGreen, and with a 561 nm laser and 610/20 filter for detection of LactC2-mCherry. Photon multiplier tube voltages were 375 V for forward scatter, 260 V for side scatter, 370 V for LactC2-mCherry, and 550 V for dsGreen detection. Loader settings were set to 50 μL injection volume with no mixing and 800 μL wash between sample runs. For each sample ~20000 events were recorded. The raw data were analyzed and pre-processed using Cytobank (https://community.cytobank.org/) to filter out possible aggregates and liposome debris by selecting the main population in the SSC vs FSC plot, and subsequently filtering out low-fluorescence events when plotted against the SSC signal. The gating strategy with fluorescence intensity thresholding is described in Supplementary Fig. 5. Text files with all SSC-A, dsGreen, and LactC2-mCherry intensity values were exported from Cytobank and plotted with MATLAB.

## Confocal microscopy

Liposome samples (2 μL) were transferred in custom-made glass chambers functionalized with 1 mg mL$^{-1}$ BSA-biotin:BSA and 1 mg mL$^{-1}$ Neutravidin, and pre-filled with 13 μL of a staining solution (1× dsGreen and 3 μM LactC2-mCherry in PB buffer). Chambers were incubated in the dark at room temperature for 1 h. In kinetic experiments (Supplementary Fig. 26), 7.5 μL of liposome suspension was mixed with 4.5 μL of dilution solution (1× PURE*frex* 2.0 solution I, 2.6× dsGreen and 22.5 μM LactC-mCherry) and loaded on a dry oleoyl-CoA lipid film as described above. The sample was incubated in the dark for 30 min at 4 °C before being transferred into a microscopy chamber and further incubated in the dark for 30 min at 4 °C to allow for liposome sinking. Confocal microscopy imaging was carried out on a Nikon Eclipse Ti (NIS-Elements AR software) using a 100× oil immersion objective. When needed, in situ incubation was performed with a heated stage at 34 °C. Laser settings for image acquisition were set to: 488-nm laser with 20 HV, -10 offset, and 1.0 intensity for dsGreen, 561-nm laser with 50 HV, -10 offset, and 1.00 intensity for LactC2-mCherry, and 640 nm laser with 95 HV, -5 offset, and 5.00 intensity for Cy5 membrane dye. Each sample was imaged by automated acquisition of 10 by 10 fields of view, stitched together with a 5% overlap. The sample height was adjusted manually to detect as many liposomes as possible, while also avoiding background debris.

## Image analysis with SMELDit

Confocal images were analyzed manually with Fiji[44] and automatically with SMELDit, an in-house-developed MATLAB script to automatically extract single liposome features while indexing each analyzed liposome. In short, the Cy5 and mCherry channels of each image are combined and convolved with a Laplacian filter kernel to determine membrane boundaries. To set what pixels belong to the inside of each liposome a binarization step, with a consistent cutoff based on previous data, followed by a filling and erosion step were utilized. The resulting binary image displayed separate segments, each representing the lumen of individual liposomes. This step was followed by an additional selection for filtering out the segments that could correspond to lipid aggregates or other noise sources. Then, all segments were analyzed individually for a circularity (C) check defined by $C = P^2/(4\pi A)$, where $P$ is the perimeter length and $A$ is the area of the segment. A 'true liposome' threshold was set for $C$ values between 0.5 and 2.0.

LactC2-mCherry aggregates were filtered out by rejecting events whenever LactC2-mCherry intensities were higher than a pre-set cutoff inside the lumen. The segments that passed these extra filtering steps were considered liposomes and were saved individually as 60 by 60-pixel cropped images with a given ID number. For each individual liposome SMELDit measures the apparent radius, average Cy5 intensity and variance on the membrane, average LactC2-mCherry intensity and variance on the membrane, and average dsGreen intensity and variance on the lumen. Per sample, SMELDit displays all single-liposome extracted data in a scatter/histogram interactive GUI on which the user can draw regions of interest (ROI) for extracting information about liposome subpopulations. Here, ROI 1–4 were defined in SMELDit using negative controls for thresholding. Finally, once ROIs are drawn, example liposomes from the ROI are retrieved as a liposome-montage. ROI liposome data can be saved separately, and ROI coordinates can also be saved and transferred to analyze another sample.

## Deep learning-based image analysis

Two image-based machine learning algorithms for the identification of liposomes were trained based on the YOLOv7-tiny model (https://github.com/DanelonLab/YOLO-for-liposomes). The first algorithm was trained for the identification of liposomes using only the membrane channel (hereon referred to as yolov7-seg). The intention behind this model was to obtain an object detection tool for the identification of liposomes with reduced dependency on sample quality and stricter discrimination criteria that can later be implemented into diverse analysis pipelines, including SMELDiT. The total pool of data consisted of 26 JPEG images of 512 by 512 pixels, containing only the membrane channel from multiple samples of varied quality and contrast settings. A single class of object, *Liposome*, was used to label objects within images using Label Studio (https://github.com/HumanSignal/label-studio), by manually drawing a bounding box tightly around the membrane of desired liposomes (Supplementary Fig. 13). Labeled liposomes were objects that had a continuous, complete circumference (thus excluding edge-cropped, partially out of plane, and stacked vesicles), single membranes (excluding liposomes with multi-vesicular contents and multilamellarity), visually assessed high circularity, and, in cases of vesicle aggregates, vesicles where most of the individual membranes could be discerned. With these considerations, a total of 2689 objects were labeled. The dataset was then randomly split in two subsets: A training subset consisted of 21 images containing 2130 liposomes, and a validation subset with five images and 559 liposomes, accounting for 20.8% of the total dataset.

Model training was performed on a Google Collab pipeline available at https://github.com/DanelonLab/YOLO-for-liposomes, using an A100 GPU. To increase the robustness of the model, image modification parameters in the.cfg training file was activated. For the case of both expose and saturation, a modifier of 1.5 factors was instructed and rotation of images during training was set to 37 degrees. Once training of the model was concluded, the model was evaluated against the validation dataset, throughout a range of confidence thresholds going from 50 to 98%. Plotting of precision vs recall at each confidence threshold (Supplementary Fig. 13), it can be seen that further reduction in confidence threshold lowly impacts recall and precision below 85%, and precision remains above 70% even at low (50%) confidence. In addition, the model was tested using four independent images of 512 by 512 pixels in which 353 "desired" objects were identified manually. Yolov7-seg was able to identify 306 objects with 90% confidence, all the way to 357 objects at 75% confidence (Supplementary Fig. 13).

A second model for phenotype identification was trained using the pipeline described above with experimental data from synthetic cell module integration and negative controls. The dataset consisted of 42 JPEG images of 512 by 512 pixels that included all the channels from the original files. Six different classes of objects were created to

include the distinct phenotypes observed in fully active synthetic cells: From classes 1 to 6 in the order *Replication, Replication condensate, Integration, Integration condensate, No phenotype*, and *PLsyn*. In total, 4469 liposomes were labeled in the samples. The training subset contained 824 class 1, 74 class 2, 228 class 3, 24 class 4, 2224 class 5, and 628 class 6 objects. Training was performed using the same parameters as for yolov7-seg. The resulting model was tested against the validation dataset that included 467 liposomes. We confirmed the algorithm is capable of identifying all phenotypes it was trained on, but it goes through faster decrease of precision upon reduction of confidence, compared to YOLOv7-seg. In addition, the maximum recall achieved was 79% at a 50% confidence threshold (Supplementary Fig. 14). However, for categories with reduced representation in the training dataset, the model is prone to miss or fails to identify objects.

## Quantitative PCR analysis

One to two microliters of liposome suspension were collected, incubated 15 min at 75 °C for DNAse I heat inactivation, and 100× diluted in MilliQ water. Ten microliter qPCR reactions were prepared with 1× PowerUP SYBR Green Master Mix (Applied Biosystems), 400 nM of each forward and reverse primer (976/977 ChD for *p2*, 980/981 ChD for *p3*, 1125/1126 ChD for *pssA*, 1119/1120 for *plsB*, 1410/1411 ChD for *plsC*, 1408/1409 ChD for *cdsA*), and 1 μL of the diluted liposome sample. Solutions were transferred to a qPCR 96-well plate (Thermo Fisher) that was sealed with an adhesive transparent film (Thermo Fisher) and spun down for 15 seconds. Measurements were performed on a Quantstudio 5 Real-Time PCR instrument (Thermo Fisher) using the protocol: 2 min at 50 °C, 5 min at 94 °C, 45 cycles of 15 s at 94 °C, 15 s at 56 °C, 30 s at 68 °C, 5 min at 68 °C, and a final melting curve stage from 65 °C to 95 °C. Sample DNA concentrations were calculated from standard curves generated using DNA templates of known concentrations ranging from 1 fM to 1 nM (9 μL of qPCR reaction + 1 μL of DNA). Data were further analyzed with QuantStudio Design and Analysis software v1.4.3 (Thermo Fisher).

## Recovery of *DNArep-PLsyn* DNA from liposomes

Liposome samples diluted 100 times in MilliQ water for qPCR measurement were utilized also for DNA PCR recovery. Reactions were set up to either amplify three fragments or a single, near full-length, fragment from the *DNArep-PLsyn* genome. Twenty to 50 μL reaction solutions were assembled in a 1× Xtreme Buffer with 300 nM of each primer (all primers and details about the corresponding PCR amplification targets can be found on Supplementary Table 2), 0.4 mM of each dNTP, 2–5 μL of the diluted liposome solution, and 0.02 U μL⁻¹ KOD DNA polymerase. The thermal cycler was programmed to follow 2 min at 94 °C for polymerase activation, and 30 cycles of (98 °C for 10 s, 60 °C for 20 s, 68 °C for 2 min for fragments A,B,C, and 5 min for one-fragment PCR-recovery). PCR products were analyzed by agarose gel electrophoresis.

## Bulk IVTT reactions

Ten microliter bulk IVTT reactions were performed with PURE*frex* 2.0 using 500 pM DNA template according to the manufacturer's guidelines (unless specified otherwise). To visualize synthesized protein products, the reaction was supplemented with 1 μL GreenLys solution (FluoroTect GreenLys, Promega) and incubated for 16 h at 37 °C. Samples were then supplemented with 1 μL of RNase A (4 mg mL⁻¹) and 1 μL of RNase One (10 U μL⁻¹), and incubated for 1-2 h at 37 °C for complete RNA digestion. 10 μL of the RNA-digested sample were mixed with 1× Laemmli sample buffer and 10 mM of DTT, final concentrations. The reaction mixtures were incubated at 95 °C for 5 min and loaded on a 12% SDS-PAGE gel that was run for 1 h at 100 V, followed by another 50–60 min at 130-160 V. GreenLys labeled proteins were visualized on a fluorescence gel imager (Typhoon, Amersham Biosciences) using Cy2 (488 nm), Cy3 (532 nm), and Cy5 (635 nm)

lasers with band-pass filters of 515-535 nm for Cy2, 560-580 nm for Cy3, and 655-685 for Cy5. Laser PMT voltages were set to 500 for Cy2 and automatic adjustment for Cy3 and Cy5. The SDS-PAGE gel was further stained overnight with Instant Blue (expedeon), destained the next day with MilliQ water, and visualized with a ChemiDoc imaging system (Bio-Rad).

## Analysis of lipid content by LC-MS

Two to four microliters of liposome suspension were diluted 10× in a sample preparation solution consisting of a 1:1:98 ratio of 0.5 M EDTA:200 mM acetylacetone:100% methanol. The solution was sonicated for 10 min, and centrifuged at 16.1 g and room temperature for 5–10 min. 15 μL of the supernatant with the soluble lipid phase were transferred to a 25 μL glass insert within a 2 mL LC-MS glass vial, further diluted 4× with the sample preparation solution, and stored under argon at −20 °C until use (less than a week). For LC-MS sample analysis, ~10 μL sample kept at 4 °C were injected into a 6460 Triple Quad LC-MS stocked with a ACQUITY UPLC Peptide CSH C18 Column with a mobile phase A of 0.05% ammonium hydroxide and 2 mM acetylacetone in water, and a mobile phase B of 80% 2-propanol, 20% acetonitrile, 0.05% ammonium hydroxide and 2 mM acetylacetone, at a flow rate of 300 μL min⁻¹ and column temperature of 60 °C. An A:B ratio of 70:30 was set to equilibrate the column. Upon sample injection, the A:B ratio was slowly changed to 100% mobile phase B and kept for 2 min. Then, the 70:30 ratio was gradually restored and kept until the end of each sample run. Transitions were established by multiple reaction monitoring[2,45]. The lipids were ionized, selected by mass in a first ion filter, fragmented by collisions with nitrogen gas in a collision cell and then the fragment masses were monitored in another ion filter. Data were analyzed with Skyline-daily. Peak areas were exported and normalized to DOPG peak areas. When two injections were done per sample, the averaged peak area was considered.

## Statistics and reproducibility

For data presented in graphs, the figure legends provide the number of biological replicates that were performed. All experiments have been reproduced at least three times (independent, biological repeats) except for some data shown in Supplementary Fig. 11, 16, 18, 19, 22, for which two repeats were performed. For the gel presented in Fig. 1e, data from two additional independent experiments are shown in Supplementary Fig. 3 (uncropped gels). Statistical tests were performed in Figs. 2e and 5f as described in the legends.

## Reporting summary

Further information on research design is available in the Nature Portfolio Reporting Summary linked to this article.

## Data availability

Lipidomics data generated in this study have been deposited in the EMBL-EBI Metabolights database under accession code MTBLS13437. Plasmid sequences have been deposited to NCBI with accession codes PX740491 to PX740496 (Supplementary Table 1 contains the embedded hyperlinks). Raw LC-MS data and plasmid maps are available on GitHub at https://github.com/DanelonLab/pMAR3 and archived on Zenodo with https://doi.org/10.5281/zenodo.18323307. The source data underlying plotted data and the uncropped/unedited scans of gel images are provided in the Source Data file. Source data are provided with this paper.

## Code availability

All the codes regarding microscopy image analysis are available on GitHub at https://github.com/DanelonLab/pMAR3 and archived on Zenodo with https://doi.org/10.5281/zenodo.18323307.

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

## Acknowledgements

The authors are thankful to Miguel de Vega and Alicia del Prado from the Centro de Biología Molecular Severo Ochoa, Madrid, for generously providing the purified SSB and DSB proteins. We also thank Adja Zoumaro-Djayoon for her support on the LC-MS lipidomic experiments, Marijn van den Brink for establishing the YOLO pipeline, and Ilja Westerlaken for her assistance in constructing *DNArep-PLsyn* genome batches. CD acknowledges funding from the Netherlands Organization for Scientific Research (NWO/OCW) through the "BaSyC, Building a Synthetic Cell" Gravitation grant (024.003.019) and from Agence Nationale de la Recherche (ANR-22-CPJ2-0091-01).

## Author contributions

C.D. acquired funding, conceived and supervised the project. A.M.R.S., F.R.G., and C.D. designed the experiments. A.M.R.S. and F.R.G. performed the experiments and analyzed the data. M.T. developed SMELDit and contributed preliminary liposome experiments. L.S.H. designed and performed the molecular cloning of plasmid pY003 and pY005 and helped with the DNA recovery and kinetics experiments. All authors discussed the results. A.M.R.S. and C.D. wrote the manuscript with input from all the authors.

## Competing interests

The authors declare no competing interests.
