## [Peer Review File · Nature Communications]

A synthetic cell with integrated DNA self-replication and lipid biosynthesis

Corresponding Author: Professor Christophe Danelon

Version 0:

Reviewer comments:

Reviewer #1

(Remarks to the Author)

This manuscript by the Danelon group presents the construction of synthetic cells integrating in vitro DNA replication and membrane biosynthesis. The authors designed a synthetic genome encoding both DNA replication proteins and membrane biosynthesis enzymes, which are expressed using the PURE system. The experimental design is thoughtful, and the data are clearly presented.

Main concerns:

While the system is functional in principle, there are two major limitations—both acknowledged by the authors (lines 281–283):

1. The majority of liposomes did not exhibit the joint phenotype of both DNA replication and phospholipid biosynthesis, indicating low co-expression efficiency. The proportion of functional synthetic cells remains limited, raising concerns about the robustness of the system.
2. Despite lipid synthesis, no detectable membrane expansion was observed. This contrasts with previous work (e.g., Exterkate et al., ACS Synth. Biol. 2018, 7, 1, 153–165), where purified enzymes enabled membrane growth. This suggests that the current system does not produce sufficient phospholipids to support physical expansion, challenging the claim of “membrane biosynthesis” in the title.

To address these concerns, the authors should better improve the system’s efficiency to overcome these limitations.

Specific comments:

1. Figure 1a: All abbreviations should be defined in the figure legend for clarity.
2. Line 115: The proportion of liposomes showing both DNA replication and phospholipid synthesis (ROI 2) is reported as less than 3%, which is particularly low. The authors should discuss strategies to increase this percentage, which is critical for realizing a functional synthetic cell.
3. Figure 3a: It is unclear whether the small panels on the right are cropped from the large panel on the left. The correspondence between the images should be clarified. Also, the border colors seem inconsistent with the rest of the manuscript. For consistency, DNA replication should be indicated in magenta and phospholipid synthesis in blue.
4. Lines 171–172: The authors note that the yield of newly synthesized lipids is insufficient for visible membrane growth. Could they estimate the minimum amount of lipids required for such growth? Adding chemically synthesized lipids might help determine this threshold and allow comparison with the lipid yield from the current system.
5. Line 180: While the DNA replication and phospholipid synthesis modules do not interfere with each other, how can their activities be maximized? Enhancing both pathways may help increase the ROI 2 population and improve system efficiency.

6. The resolution of all supplementary figures is too low. For example, plasmid maps are not legible. Please provide high-resolution versions in the supplementary information.

7. Figure S1b legend: The phrase “without (left) and without (right)” appears to be an error. Please revise for clarity.

(Remarks on code availability)

Reviewer #2

(Remarks to the Author)

This manuscript presents a synthetic DNA genome that drives both self-replication and membrane phospholipid biosynthesis within liposomes. The authors express six genes from a linear construct—two encoding the DNA replication machinery and four involved in phospholipid synthesis—using the PURE system. The genome is capable of self-replication, and the encoded enzymes catalyze the production of DOPS (1,2-dioleoyl-sn-glycero-3-phospho-L-serine), a key phospholipid intermediate. These processes are reconstituted in liposomes, establishing a platform for studying the integration of core cellular functions in a minimal synthetic context.

The combination of confocal microscopy and computational analysis is a key strength of the study. The authors quantify single-vesicle activity using their custom software SMELDit (<https://github.com/DanelonLab/SMELDit>), which extracts fluorescence intensity measurements for DNA replication (dsGreen) and membrane biosynthesis (LactC2-mCherry). However, the current classification strategy—based on fixed intensity thresholds and a quadrant-based map dividing vesicles into four phenotypes—is limited. It imposes artificial boundaries that may not reflect the true distribution of phenotypic states and is highly sensitive to variability in fluorescence intensity across experiments.

A more robust and unsupervised strategy—such as dimensionality reduction using UMAP (Uniform Manifold Approximation and Projection) followed by density-based clustering (e.g., HDBSCAN) would enable phenotype identification based on the structure of the data itself, without imposing predefined categories. This would improve reproducibility and better account for biological variability and differences across experimental replicates, avoiding arbitrary thresholds.

Moreover, the imaging dataset contains rich spatial and morphological information that remains underexploited. The current analysis appears to rely primarily on average fluorescence intensities, but additional features such as vesicle area, circularity, membrane texture, internal intensity variance, and the presence of localized replication foci could provide a more complete description of vesicle states. These features could then be integrated into the clustering pipeline, enhancing the ability to resolve subtle or intermediate phenotypes and to explore relationships between genotype, phenotype, and compartmental architecture.

In addition to rethinking the classification approach, image segmentation itself could benefit from modern deep learning-based methods, which have shown superior performance in detecting and segmenting biological compartments, particularly in heterogeneous and noisy microscopy data. Neural network architectures trained on a representative annotated dataset would likely improve segmentation accuracy, reduce bias from manual parameter tuning, and allow for better generalization across experiments.

In terms of implementation, while the availability of the SMELDit code is appreciated, the fact that it relies on MATLAB—an expensive, engineer-oriented commercial platform—limits accessibility and reuse, especially among researchers in the life sciences. Reimplementing the pipeline using an open-source, community-supported language such as Python (e.g., with napari, or PyTorch) or Java (e.g., Fiji/ImageJ) would significantly increase its utility and adoption.

Finally, since the GitHub repository does not include example datasets or segmentation outputs due to file size constraints, I encourage the authors to provide at least one representative dataset—including raw images, segmentation masks, and extracted features—to support reproducibility and allow others to test and extend the tool.

In summary, this is a valuable and well-executed study that advances synthetic cell engineering by integrating genome-driven replication and membrane biosynthesis within liposomes. The imaging data is central to the conclusions of the manuscript. Incorporating more advanced computational methods for segmentation, feature extraction, and phenotype classification would enhance the reproducibility and interpretability of the analysis, ultimately increasing the impact of the work.

(Remarks on code availability)

The authors provide a link to their image analysis code SMELDit on GitHub (<https://github.com/DanelonLab/SMELDit>), which includes a README file with basic usage instructions. The documentation outlines the general pipeline and required input files.

I have not attempted to install or run the code, as no example datasets or sample images are currently provided. The repository states that image data and segmentation outputs are not included due to size limitations. I have requested that the authors provide a representative dataset to facilitate testing and reproducibility.

In its current state, the code appears potentially useful for the community, but usability and reproducibility are limited by the absence of test data and by the reliance on MATLAB, which is a commercial platform not easily accessible to all researchers. Porting the tool to an open-source language such as Python or Java would make it significantly more accessible and reusable.

Reviewer #3

(Remarks to the Author)

Sierra et al. describe the integration of two fundamental modules likely required to achieve the development of a synthetic cell, namely DNA replication and lipid synthesis (metabolism). They demonstrate that by combining PURE with a linear template encoding 6 proteins (2 required for DNA replication, and 4 enzymes required for lipid synthesis) and encapsulating the reaction in liposomes. Using FACS and direct imaging of these liposomes, they were able to show that both DNA replication and lipid synthesis occur in these liposomes, although the co-occurrence of these two processes was a surprisingly rare event compared to each module alone. The FACS data has some shortcomings in regards to demonstrating that DNA replication occurred, although these shortcomings are for the most part resolved with the direct imaging approach. Although it might be worthwhile to consider conducting a time-course experiment if possible. Overall, the manuscript is well written, and for the most part well supported by data (with the exception of Fig 2 and corresponding Supp figures). The authors should address these issues in a revised document. Overall, this manuscript demonstrates a significant step forward which will be of general interest to the field.

Specific comments:

There appears to be a copy – paste mistake in Figure S5 (page 6) as the two figures in the last two rows – middle column are identical.

Although PS synthesis seems to be fairly robust and relatively clearly visible based on the FACS data provided in Figure S5, whether DNA replication was functional is much less clear / apparent. For example, the seemingly most obvious DNA replication event occurred in replicate 2 at 34C, but the confounding factor is the fact that the positive control data points are in general about one order of magnitude higher than the negative control. Could this be due to simply having generally higher DNA concentrations from the beginning in the positive control liposomes, or is the expectation that all liposomes are expected to perform DNA replication. The analysis on the other hand (using a threshold) suggests that the expectation is that only a fraction of liposomes will be able to perform functional DNA replication. Furthermore, in replicate 6 at 34C it appears that the threshold for the NC and PC is not the same! In general, the evidence for DNA replication based on Figure S5 seems relatively weak. Would it be possible to perform a FACS analysis of the same samples both at T=0 and at 16 hours, by subjecting a fraction of the reaction to FACS at the beginning of the reaction. This would provide a good baseline for identifying whether the liposome populations shift in the FACS measurement relative to T0.

The data shown in Fig 2e should be supported by a statistical test to indicate whether or not there are indeed significant differences in DNA replication observed. Since this qPCR is a bulk assay it likely can't exclude the possibility that DNA replication, if it occurs, actually occurs inside liposomes or outside of liposomes (liposomes could burst and release their content and DNA replication could occur in the supernatant)?

Direct imaging of liposomes as shown in Fig 3 is more convincing than the FACS data in regards to DNA replication, although here it would be good to provide a time-course as well (image liposomes at T=0 and then repeat imagine at reasonable intervals of every few hours). The following experiments in Fig 4 and 5 also nicely add to the evidence that DNA replication indeed occurs in liposomes. Could it therefore be that FACS is maybe not an ideal method for characterizing these liposomes as they could rupture and disintegrate in the FACS for example?

It remains somewhat curious why such a low fraction of liposomes are exhibiting DNA replication as well as liposome synthesis...

(Remarks on code availability)

Version 1:

Reviewer comments:

Reviewer #1

(Remarks to the Author)

The authors adequately addressed my comments.

(Remarks on code availability)

Reviewer #2

(Remarks to the Author)

The authors have substantially improved their manuscript "A synthetic cell with integrated DNA self-replication and lipid biosynthesis." The revisions directly and thoroughly address the reviewers' previous concerns. The new experiments, clearer data presentation, and expanded discussion all strengthen the manuscript and support its conclusions.

The additional work using a more stable plasmid-derived DNA template convincingly improves the system's efficiency and reproducibility, leading to a significantly higher fraction of functional synthetic cells. This effectively addresses one of the main limitations noted earlier. Furthermore, the implementation of a deep-learning-based image analysis pipeline and the public release of both SMELDit and the new code on GitHub enhance the transparency, reproducibility, and community value of the study.

The revised Discussion provides a balanced perspective on the current capabilities and limitations of the system, including the remaining challenge of achieving membrane expansion. The authors now place their findings appropriately within the broader context of synthetic biology and cite relevant prior work. The methodological details and data analyses appear sound and sufficiently detailed to allow reproduction of the work.

The manuscript now presents a solid and well-supported contribution to the field of bottom-up synthetic biology. It demonstrates the successful integration of DNA self-replication and lipid biosynthesis within synthetic cells and provides useful tools for further exploration of minimal cell systems. I am satisfied with the revisions made and have no further major concerns.

(Remarks on code availability)

I have not directly reviewed or tested the code. However, the authors have made their custom software and neural network pipelines openly available through GitHub repositories (SMELDit, pySMELDit, and YOLO-for-liposomes). The inclusion of README files and example data, as described in the manuscript and rebuttal, suggests that the code is documented and accessible to the community. The open availability of these tools represents a valuable step toward transparency and reproducibility, even though I did not independently verify their functionality.

Reviewer #3

(Remarks to the Author)

The authors addressed all of my suggestions in this revision and I recommend the manuscript for publication.

(Remarks on code availability)

Rebuttal letter for:

Manuscript: "A synthetic cell with integrated DNA replication and membrane biosynthesis"
by Restrepo Sierra et al.

Article reference: NCOMMS-25-21548

We thank the reviewers for recognizing the importance of our work and for their constructive comments that helped us improve the manuscript. The Referee reports are in **black text** and our point-by-point responses are in **blue text**. Changes in the manuscript (main text and Supplementary information) are highlighted in **red text**.

We are pleased to provide you with a substantially revised manuscript that addresses all referees' concerns. In particular:

- a) We performed additional experiments with a more robust source of DNA and managed to significantly improve the efficiency of the system.
- b) We fully engaged with the guidance from reviewer #2 and made our custom software SMELDit broadly accessible on Python. The code and related files are available on GitHub.
- c) We used a neural network-based image analysis pipeline for segmentation and phenotype classification and made the code and related files available on GitHub.

REVIEWER COMMENTS

Reviewer #1 (Remarks to the Author):

This manuscript by the Danelon group presents the construction of synthetic cells integrating in vitro DNA replication and membrane biosynthesis. The authors designed a synthetic genome encoding both DNA replication proteins and membrane biosynthesis enzymes, which are expressed using the PURE system. The experimental design is thoughtful, and the data are clearly presented.

Reply: We thank the reviewer for these positive comments.

Main concerns:

While the system is functional in principle, there are two major limitations—both acknowledged by the authors (lines 281–283):

1. The majority of liposomes did not exhibit the joint phenotype of both DNA replication and phospholipid biosynthesis, indicating low co-expression efficiency. The proportion of functional synthetic cells remains limited, raising concerns about the robustness of the system.

Reply: About 8% of the total liposomes exhibit the joint phenotype (e.g., Fig. 3b). Although this clearly shows room for improvement, more than hundred vesicles with both module activities are routinely obtained in a sample of 2 μ L and in a single imaging z-plane (see Movie). This would mean that we likely produce thousands of active vesicles with the joint phenotype in a full reaction of 10-20 μ L. Moreover, we cannot exclude that liposomes assigned as ‘inactive’ based on fluorescence intensity thresholding would present some basal activity that is below the detection sensitivity.

Nonetheless, we agree that the fraction (not per se the absolute number) of functional synthetic cells is rather low, suggesting that some factors limit the system’s efficiency.

In the revised manuscript, we discussed the current limitations of the system and experimentally tested a hypothesis, which significantly improved robustness (see below).

We extended the first paragraph of the Discussion as follows:

“Although we routinely obtained more than a hundred vesicles exhibiting the joint phenotype per sample, these only represent about 8% of the total liposome population when using a DNA template generated by overlapping PCR. We suspected that the robustness of the system may be limited by the DNA quality and purity. Using a stable source of DNA, namely a sequence-verified circular plasmid to generate the linear DNAREP-PLsyn template, improved the system’s efficiency over a factor of 3.5 with ~30% of fully functional synthetic cells. We attribute this improvement to a higher purity of the full-length DNA that also enables to work with increased concentrations of active template. The remaining fraction of nonfunctional synthetic cells probably arises from deficient molecular compositions inside vesicles, which cannot support all functions concurrently. For instance, differences in the loading of DNA or translation machinery between liposomes probably lead to varying levels of expressed genes, while uneven supply of substrates or cofactors for DNAREP and PLsyn would result in varying outputs of the modules. Even if it is clear that prototype synthetic cells will not have the level of control and robustness of natural living cells, a major challenge will be to mitigate this heterogeneity and increase the proportion of active vesicles.”

2. Despite lipid synthesis, no detectable membrane expansion was observed. This contrasts with previous work (e.g., Exterkate et al., ACS Synth. Biol. 2018, 7, 1, 153–165), where purified enzymes enabled membrane growth. This suggests that the current system does not produce sufficient phospholipids to support physical expansion, challenging the claim of “membrane biosynthesis” in the title.

Reply: The current system does not produce sufficient lipids to support visual expansion of the membrane. In the Discussion we wrote: “*challenges remain to achieve physical growth of liposomes upon lipid biosynthesis*”.

In the study of Exterkate et al. (new ref. 25), the authors have reconstituted detergent-solubilized enzymes into small/large vesicles, showing nearly stoichiometric conversion of fatty acid substrates into phospholipids. Reactions were not compartmentalized in liposomes and the enzymes were not genetically encoded. De novo membrane synthesis and vesicle growth were also demonstrated by the group of Devaraj (new refs. 26, 27). Here too, the experimental framework radically differs from ours, and the respective advantages and shortcomings are different.

We agree that these previous studies should be cited as they show that alternative methods for lipid production exist and may be coupled to gene expression with PURE system and DNA replication. Actually, we attempted to produce phospholipid analogues by native chemical ligation, as developed in ref. 26. The precursors sodium 2-mercaptoethane-sulfonate (MESNA) oleate, 1-oleoyl-2-(L-Cys)-*sn*-glycero-3-phosphocholine (o-LPC), and 1-palmitoyl-2-(L-Cys)-*sn*-glycero-3-phosphocholine (p-LPC) were kindly provided by the Devaraj lab. Unfortunately, the reactive precursors severely inhibited gene expression in PURE system and we decided to not pursue this strategy for now. We also tried to utilize oleic acid as a phospholipid precursor by extending the Kennedy pathway upstream with the FadD enzyme, but phospholipid synthesis yield (here DOPE) was not significantly higher than with oleoyl-CoA despite a 10-fold higher starting concentration (PhD thesis Duco Blanken, link: <https://repository.tudelft.nl/record/uuid:363f4643-0a68-4726-9945-e8daf6e0350c>)

Accordingly, we added in the Discussion: “While the DNA amplification fold may be sufficient for the partitioning of the genetic material amongst daughter vesicles upon division, the bottleneck for a cell cycle is on the yield of synthesized phospholipids. With the produced DOPS representing only <7% of the total lipid amount, the current system cannot support physical expansion of the membrane. Using purified, detergent-solubilized enzymes reconstituted into large vesicles, Exterkate et al. showed efficient conversion of the fatty acid precursor, which enabled membrane growth [25]. In other studies, de novo membrane synthesis was demonstrated with phospholipid analogues produced by native chemical ligation [26,27]. The compatibility of these approaches with PURE system driving the expression of an internal DNA program remains to be investigated.”

In our opinion, the fact that phospholipids are synthesized and stably insert in the membrane does not conflict with ‘membrane biosynthesis’ in the title. We acknowledge however that the term might be confounding and we changed ‘membrane’ to ‘lipid’ in the new title. The fact that physical expansion was not detected shows limitations that are more extensively discussed in the revised manuscript.

To address these concerns, the authors should better improve the system's efficiency to overcome these limitations.

Reply: We engaged with the suggestion of the reviewer and performed additional experiments to overcome some of the limitations. We managed to improve the system's efficiency by constructing a circular plasmid as a stable source of DNA to generate the linear *DNArep-PLsyn* genome instead of overlapping PCR. Expression of 1 nM of this new template in liposomes resulted in 30% of liposomes exhibiting the joint phenotype, which represents an improvement of 3.5 times.

In the Results section, line 331, we added a paragraph and included four new supplementary figures reporting these findings:

“...DNA libraries. High amounts of the PCR-amplified genome could be obtained, allowing for nanomolar input concentrations. Its expression in liposomes improved the robustness and efficiency of the system, yielding ~30% (vs 8%) of fully active synthetic cells and over 37% (vs 17%) with at least one functional module, while reducing variability between replicates (Supplementary Figs. 23-25). The change of phenotype map compared to Fig. 3c (template obtained by overlapping PCR) suggests an increased ability of DNArep-active liposomes to also produce phospholipids. Time-lapse imaging of phenotype development shows that DNArep precedes PLsyn and the activity of both modules reaches a plateau after 6 hours (Supplementary Fig. 26).”

Specific comments:

1. Figure 1a: All abbreviations should be defined in the figure legend for clarity.

Reply: We defined all abbreviations in the revised manuscript.

2. Line 115: The proportion of liposomes showing both DNA replication and phospholipid synthesis (ROI 2) is reported as less than 3%, which is particularly low. The authors should discuss strategies to increase this percentage, which is critical for realizing a functional synthetic cell.

Reply: The exact percentage of liposomes detected in ROI 2 varies from one biological repeat to another, but is generally lower than 10%. In the revised manuscript, we extensively discussed the current limitations and reported new results showing an improved efficiency with ~30% liposomes in ROI 2. See our replies to the comments above.

3. Figure 3a: It is unclear whether the small panels on the right are cropped from the large panel on the left. The correspondence between the images should be clarified. Also, the border

colors seem inconsistent with the rest of the manuscript. For consistency, DNA replication should be indicated in magenta and phospholipid synthesis in blue.

Reply: We accidentally swapped the frame colors of 'DNA replication' and 'PS synthesis' in the large panel. Thank you for pointing this out. For clarity, we added in the figure legend: "The cropped images correspond to the framed liposomes in the large panel."

4. Lines 171–172: The authors note that the yield of newly synthesized lipids is insufficient for visible membrane growth. Could they estimate the minimum amount of lipids required for such growth? Adding chemically synthesized lipids might help determine this threshold and allow comparison with the lipid yield from the current system.

Reply: We estimate that an increase of the liposome diameter of 20% would be required to unambiguously observe membrane growth. The microscope resolution (around 250 nm) theoretically enables smaller changes to be detected but tracking of single vesicles over multiple time frames may result in z-plane changes, which eventually can alter length measurements. Moreover, FRET-based assays could be used to monitor smaller changes but the present considerations pertain only to direct visualization of liposome growth.

For a 5- μm diameter liposome undergoing a 20% increase in size ($\varnothing = 5 \rightarrow 6 \mu\text{m}$), the corresponding excess membrane surface area is $34.6 \mu\text{m}^2$ ($A = 113.1 - 78.5 \mu\text{m}^2$, i.e., 44% increase). Assuming a lipid molecular surface area of 70 \AA^2 (corresponding to DOPC, the most abundant lipid species in our vesicles), the additional lipid amount required is 98.7×10^6 lipids per vesicle.

At the sample level, in the ideal case scenario, where all oleoyl-CoA (176 μM) is consumed and fully converted into DOPS, 88 μM of phospholipids are synthesized (DOPS is a two-acyl chain lipid). Given a total concentration of lipids of 2 mM when starting genome expression, and under the assumptions i-iii reported below, one can calculate that the liposome surface area increases by 4.4% (0.088 mM DOPS is 4.4% of 2 mM total lipids and the membrane surface area scales linearly with the number of lipids).

Assumptions:

- (i) lipid synthesis is homogeneously distributed across all liposomes
- (ii) all synthesized lipids assemble into the membrane
- (iii) all lipid species in the membrane have the same molecular surface area.

As the diameter scales with the square root of the area, the liposome size increases by 2.1%. For a liposome diameter of 5 μm , this means a diameter increase of 105 nm, which is below the detection limit of our light microscope and much lower than a vesicle growth of 1 μm for unambiguous observation.

The concentration of O-CoA that needs to be converted into DOPS to support an increase of 1 μm in diameter (i.e., a surface increase of 44%) would be 1.76 mM ($2 \times 880 \mu\text{M}$ DOPS), which largely exceeds the solubility of O-CoA in PURE system (estimated to be $<200 \mu\text{M}$ given its precipitation with magnesium ions) and may cause inhibition to lipid synthesis

(<https://pubmed.ncbi.nlm.nih.gov/7018587/>). Continuous addition of smaller concentrations of acyl-CoA may be possible but will require to design a dedicated flow-cell imaging chamber. Finally, we note that to sustain an increase of the liposome surface of 44%, DOPS will likely have to be converted into PE by introducing a downstream enzyme of the Kennedy pathway (Psd) to balance the surface charge density.

Adding chemically synthesized lipids (see our reply to Comment #2) or experiments of SUV/LUV fusion may indeed help determine the threshold of extra lipids that is required to observe growth in our conditions.

5. Line 180: While the DNA replication and phospholipid synthesis modules do not interfere with each other, how can their activities be maximized? Enhancing both pathways may help increase the ROI 2 population and improve system efficiency.

Reply: The new results presented in Supplementary Figs. 23-26 demonstrate that the system's efficiency can be improved with a higher concentration of a DNA genome of higher quality/purity. This was achieved by using a yeast-assembled plasmid as a PCR template to generate the linear genome instead of the one produced by overlapping PCR. Strategies for further improvement of module integration are discussed in the revised manuscript. Please see also our reply to Comment #2.

In the original Discussion we proposed to optimize the metabolic balance and resource allocation for gene expression, to introduce temporal control over gene expression, and to evolve the gene-encoded proteins.

6. The resolution of all supplementary figures is too low. For example, plasmid maps are not legible. Please provide high-resolution versions in the supplementary information.

Reply: Resolution is good enough on our PDF file. We think resolution may be reduced when uploading the file on the submission portal. We uploaded a new Supplementary Information file with (hopefully) higher resolution. Moreover, we now provide a link to access plasmid maps on Github, link: <https://github.com/DanelonLab/pMAR3/tree/DNA-sequences>

7. Figure S1b legend: The phrase “without (left) and without (right)” appears to be an error. Please revise for clarity.

Reply: We corrected as “*without (left) and with (right)*”

Reviewer #2 (Remarks to the Author):

This manuscript presents a synthetic DNA genome that drives both self-replication and membrane phospholipid biosynthesis within liposomes. The authors express six genes from a linear construct—two encoding the DNA replication machinery and four involved in phospholipid synthesis—using the PURE system. The genome is capable of self-replication, and the encoded enzymes catalyze the production of DOPS (1,2-dioleoyl-sn-glycero-3-phospho-L-serine), a key phospholipid intermediate. These processes are reconstituted in liposomes, establishing a platform for studying the integration of core cellular functions in a minimal synthetic context.

The combination of confocal microscopy and computational analysis is a key strength of the study. The authors quantify single-vesicle activity using their custom software SMELDit (<https://github.com/DanelonLab/SMELDit>), which extracts fluorescence intensity measurements for DNA replication (dsGreen) and membrane biosynthesis (LactC2-mCherry). However, the current classification strategy—based on fixed intensity thresholds and a quadrant-based map dividing vesicles into four phenotypes—is limited. It imposes artificial boundaries that may not reflect the true distribution of phenotypic states and is highly sensitive to variability in fluorescence intensity across experiments.

Reply: We generally agree that classifying phenotypes merely using fluorescence intensity thresholds is somehow limiting. Yet, it is a good method to assess the activity of the two modules in a quantitative way. As described below, we also used a machine learning-based method to segment images and identify liposomes with features that go beyond intensity thresholds. The code is available on Github: <https://github.com/DanelonLab/YOLO-for-liposomes>

Note that for each independent experimental condition a corresponding negative control was performed for each module. In each independent repeat, these negative controls were used to generate the regions of interest for each phenotype. We observed that throughout the different repeats, the thresholds could successfully be extrapolated. Then, when merging all experiments (as shown for instance in Fig 3b), we could gate the ROIs as effectively as when done individually.

Although the classification of phenotypes relies on average fluorescence intensity values, SMELDit can easily be upgraded to also extract other physical parameters. We modified the code to store and retrieve the perimeter, circularity, and variance of the Cy5 signal (to further facilitate the cleaning of data) for each individual liposome detected.

A more robust and unsupervised strategy – such as dimensionality reduction using UMAP (Uniform Manifold Approximation and Projection) followed by density-based clustering (e.g.,

HDBSCAN) would enable phenotype identification based on the structure of the data itself, without imposing predefined categories. This would improve reproducibility and better account for biological variability and differences across experimental replicates, avoiding arbitrary thresholds.

Reply: Thresholding is not arbitrary. As indicated above, thresholds are defined based on fluorescence intensity of negative controls. To note, SMELDit is not restricted to phenotype identification; it also enables quantitative analysis by extracting physical parameters, such as fluorescence intensity, variance, and circularity.

In the revised manuscript, we used a neural network-based image analysis pipeline to classify liposomes based on multimodal features that go beyond fluorescence intensity thresholding. The approach remains 'supervised' as we train the model with two/three sets of liposome images predefined by the user as 'joint phenotype', DNAREP, or PLSyn.

To inform the reader about the existence of unsupervised strategy, we added in the Discussion the following sentence and new reference 42: "Finally, incorporating more advanced computational methods, like unsupervised learning [42], for segmentation, feature extraction, and phenotype classification would enhance reproducibility of the analysis, while expanding the discovery of novel properties."

Moreover, the imaging dataset contains rich spatial and morphological information that remains underexploited. The current analysis appears to rely primarily on average fluorescence intensities, but additional features such as vesicle area, circularity, membrane texture, internal intensity variance, and the presence of localized replication foci could provide a more complete description of vesicle states. These features could then be integrated into the clustering pipeline, enhancing the ability to resolve subtle or intermediate phenotypes and to explore relationships between genotype, phenotype, and compartmental architecture.

Reply: Although liposome classification with SMELDit relied on average fluorescence intensity values, additional features, such as vesicle area and radius, circularity, intensity variances in all three channels could also be extracted to provide a more complete description of the liposome states. For instance, the lumen intensity variance of dsGreen signal could be used to identify vesicles with localized replication foci.

We added some information in the *Customization* section of the pySMELDit Readme file. For implementing additional features or improving liposome recognition, one can simply switch out the `image_analysis` function inside `image_analysis_module.py` for whatever new algorithm. Just make sure it saves the same metrics to its data file. Additional metrics to be displayed in the SMELDit GUI can be manually added to the `self.metric_names` dict in `SMELDit.py`.

As detailed in our reply to the next comment, we developed deep-learning-based methods for image segmentation and phenotype classification.

In addition to rethinking the classification approach, image segmentation itself could benefit from modern deep learning-based methods, which have shown superior performance in detecting and segmenting biological compartments, particularly in heterogeneous and noisy microscopy data. Neural network architectures trained on a representative annotated dataset would likely improve segmentation accuracy, reduce bias from manual parameter tuning, and allow for better generalization across experiments.

Reply: We have trained a YOLOv7-tiny algorithm for the detection of vesicles and phenotype identification. This machine learning algorithm recognizes liposomes by their membrane channel and generates an output file including the object classification, a detection confidence and the coordinates of a bounding box spatially delimiting the object. We made the code available on GitHub at <https://github.com/DanelonLab/YOLO-for-liposomes>, and included a README file and two subfolders with example images.

We added in Methods a new section “Deep learning-based image analysis”:

“Two image-based machine learning algorithms for the identification of liposomes were trained based on the YOLOv7-tiny model (<https://github.com/DanelonLab/YOLO-for-liposomes>). The first algorithm was trained for the identification of liposomes using only the membrane channel (hereon referred to as yolov7-seg). The intention behind this model was to obtain an object detection tool for the identification of liposomes with reduced dependency on sample quality and stricter discrimination criteria that can later be implemented into diverse analysis pipelines, including SMELDiT. The total pool of data consisted of 26 JPEG images of 512 by 512 pixels, containing only the membrane channel from multiple samples of varied quality and contrast settings. A single class of object, *Liposome*, was used to label objects within images using Label Studio (<https://github.com/HumanSignal/label-studio>), by manually drawing a bounding box tightly around the membrane of desired liposomes (Supplementary Fig. 13). Labelled liposomes were objects that had a continuous, complete circumference (thus excluding edge-cropped, partially out of plane, and stacked vesicles), single membranes (excluding liposomes with multi-vesicular contents and multilamellarity), visually-assessed high circularity, and, in cases of vesicle aggregates, vesicles where most of the individual membranes could be discerned. With these considerations, a total of 2,689 objects were labelled. The dataset was then randomly split in two subsets: A training subset consisted of 21 images containing 2,130 liposomes, and a validation subset with 5 images and 559 liposomes, accounting for 20.8% of the total dataset.

Model training was performed on a Google Collab pipeline available at <https://github.com/DanelonLab/YOLO-for-liposomes>, using an A100 GPU. To increase the robustness of the model, image modification parameters in the .cfg training file was activated. For the case of both expose and saturation, a modifier of 1.5 factors was instructed and rotation of images during training was set to 37 degrees. Once training of the model was concluded, the

model was evaluated against the validation dataset, throughout a range of confidence thresholds going from 50% until 98%. Plotting of precision vs recall at each confidence threshold (Supplementary Fig. 13), it can be seen that further reduction in confidence threshold lowly impacts recall and precision below 85%, and precision remains above 70% even at low (50%) confidence. In addition, the model was tested using 4 independent 512 by 512 pixels in which 353 “desired” objects were identified manually. Yolov7-seg was able to identify 306 objects with 90% confidence, all the way to 357 objects at 75% confidence (Supplementary Fig. 13).”

Fig. S13. Deep learning-assisted liposome identification. YOLO detection output with 80% confidence (b) vs manual selection of desired vesicles (a). Overall, most liposomes were correctly identified, although a tendency for the model to label cropped liposomes remains, as well as reduced ability to recognize small liposomes. c) Precision vs Recall plot for the model validation at varied confidence thresholds.

“A second model for phenotype identification was trained using the previously described pipeline with experimental data from synthetic cell module integration and negative controls. The dataset consisted of 42 JPEG images of 512 by 512 pixels that included all the channels

from the original files. Six different classes of objects were created to include the distinct phenotypes observed in fully active synthetic cells: From classes 1 to 6 in the order *Replication*, *Replication condensate*, *Integration*, *Integration condensate*, *No phenotype*, and *PLsyn*. In total, 4,469 liposomes were labelled in the samples. The training subset contained 824 class 1, 74 class 2, 228 class 3, 24 class 4, 2,224 class 5, and 628 class 6. Training was performed using the same parameters as for yolov7-seg. The resulting model was tested against the validation dataset that included 467 liposomes. We confirmed the algorithm is capable of identifying all phenotypes it was trained on, but it goes through faster decrease of precision upon reduction of confidence, compared to YOLOv7-seg. In addition, the maximum recall achieved was 79% at a 50% confidence threshold (Supplementary Fig. 14). However, for categories with reduced representation in the training dataset, the model is prone to miss- or fail to identify objects.”

Fig. S14. Deep learning-assisted phenotype classification. a) Precision vs Recall plot for the model validation at varied confidence thresholds. **b)** YOLO detection output with 80% confidence.

We summarized the main findings in the revised manuscript, line 219, as: “For better generalization of image analysis across experiments and sample types, we developed a deep-learning-based method for automated vesicle segmentation and phenotype classification. The model was able to identify 306 liposomes with 90% confidence out of 353 manually identified vesicles (Supplementary Fig. 13). Moreover, the model enabled classification of the phenotypes, including vesicles with localized replication foci, without relying on fluorescence intensity thresholding or other predefined parameters (Supplementary Fig. 14).”

In terms of implementation, while the availability of the SMELDit code is appreciated, the fact

that it relies on MATLAB—an expensive, engineer-oriented commercial platform—limits accessibility and reuse, especially among researchers in the life sciences. Reimplementing the pipeline using an open-source, community-supported language such as Python (e.g., with napari, or PyTorch) or Java (e.g., Fiji/ImageJ) would significantly increase its utility and adoption.

Reply: We agree with the reviewer and converted the SMELDit code to Python: pySMELDit.

Technical note: The MATLAB function called **imfill** which we used for processing grayscale images has no disclosed algorithm in the Mathworks documentation. Therefore, when converting the code to Python, we had to approximate its function with **skimage.morphology.reconstruction** which performs similarly, but not exactly the same. Therefore, the results obtained with the MATLAB and Python versions may not be identical since the liposome recognition function is slightly different.

We made the complete code of pySMELDit with example image ('test sample') available on GitHub repository and provided the new link in the Code Availability statement.

Finally, since the GitHub repository does not include example datasets or segmentation outputs due to file size constraints, I encourage the authors to provide at least one representative dataset—including raw images, segmentation masks, and extracted features—to support reproducibility and allow others to test and extend the tool.

Reply: We provided a TestCase image. Instructions on how to run the code can be found in the README file. We updated the nr-software policy accordingly.

Here is what it looks like to run:

In summary, this is a valuable and well-executed study that advances synthetic cell engineering by integrating genome-driven replication and membrane biosynthesis within liposomes. The imaging data is central to the conclusions of the manuscript. Incorporating more advanced computational methods for segmentation, feature extraction, and phenotype classification would enhance the reproducibility and interpretability of the analysis, ultimately increasing the impact of the work.

Reviewer #2 (Remarks on code availability):

The authors provide a link to their image analysis code SMELDit on GitHub (<https://github.com/DanelonLab/SMELDit>), which includes a README file with basic usage instructions. The documentation outlines the general pipeline and required input files.

I have not attempted to install or run the code, as no example datasets or sample images are currently provided. The repository states that image data and segmentation outputs are not included due to size limitations. I have requested that the authors provide a representative dataset to facilitate testing and reproducibility.

In its current state, the code appears potentially useful for the community, but usability and reproducibility are limited by the absence of test data and by the reliance on MATLAB, which is a commercial platform not easily accessible to all researchers. Porting the tool to an open-source language such as Python or Java would make it significantly more accessible and

reusable.

Reply: We fully engaged with the reviewer's suggestions and provided two versions of SMELDiT, one on MATLAB and one on Python. We also provided a representative dataset to facilitate testing.

Reviewer #3 (Remarks to the Author):

Sierra et al. describe the integration of two fundamental modules likely required to achieve the development of a synthetic cell, namely DNA replication and lipid synthesis (metabolism). They demonstrate that by combining PURE with a linear template encoding 6 proteins (2 required for DNA replication, and 4 enzymes required for lipid synthesis) and encapsulating the reaction in liposomes. Using FACS and direct imaging of these liposomes, they were able to show that both DNA replication and lipid synthesis occur in these liposomes, although the co-occurrence of these two processes was a surprisingly rare event compared to each module alone. The FACS data has some shortcomings in regards to demonstrating that DNA replication occurred, although these shortcomings are for the most part resolved with the direct imaging approach. Although it might be worthwhile to consider conducting a time-course experiment if possible. Overall, the manuscript is well written, and for the most part well supported by data (with the exception of Fig 2 and corresponding Supp figures). The authors should address these issues in a revised document. Overall, this manuscript demonstrates a significant step forward which will be of general interest to the field.

Reply: We thank the reviewer for the overall positive evaluation and constructive comments. In the revised manuscript, we conducted a time-course confocal imaging experiment and addressed the reviewer's issue about Fig. 2 and the associated Supplementary figures.

Specific comments:

There appears to be a copy – paste mistake in Figure S5 (page 6) as the two figures in the last two rows – middle column are identical.

Reply: The reviewer is correct; we made a mistake when pasting one of the graphs. This is corrected.

Although PS synthesis seems to be fairly robust and relatively clearly visible based on the FACS data provided in Figure S5, whether DNA replication was functional is much less clear / apparent. For example, the seemingly most obvious DNA replication event occurred in replicate 2 at 34C, but the confounding factor is the fact that the positive control data points are in general about one order of magnitude higher than the negative control. Could this be due to

simply having generally higher DNA concentrations from the beginning in the positive control liposomes, or is the expectation that all liposomes are expected to perform DNA replication. The analysis on the other hand (using a threshold) suggests that the expectation is that only a fraction of liposomes will be able to perform functional DNA replication.

Reply: Yes, only a fraction of liposomes was able to perform functional DNA replication. The percentage of liposomes that perform replication corresponds to % in ROI1 + ROI2 = 5.9 + 11.8 = 17.7% for replicate 2 at 34 °C. Positive data points are not one order of magnitude higher than the negative control. We noticed that in some graphs the y-axis labels were slightly shifted relative to the ticks; this may have been misleading. This layout issue is fixed.

In Fig. S5, negative control samples correspond to time-zero samples for each condition. For clarity we added a sentence in the legend of the figure. Therefore, DNA concentration from the beginning cannot be higher in the positive control liposomes. Moreover, we verified that the initial amount of DNA in positive samples did not lead to a higher dsGreen signal in the absence of replication. This can be seen in Fig. S21 (formerly S19) when comparing the dsGreen intensity at time zero (with DNA but no replication) and in the no-DNA control reaction (Fig. S5). In the course of replication, a fraction of liposomes exhibits dsGreen signal that exceeds the intensity threshold (Fig. S21a). This result clearly shows that DNA replication is functional in some liposomes (ROI1 + ROI2). Further support is provided with microscopy data.

Furthermore, in replicate 6 at 34C it appears that the threshold for the NC and PC is not the same!

Reply: The thresholds for both conditions were the same but the y-axis labels were slightly shifted from the ticks. We corrected it.

In general, the evidence for DNA replication based on Figure S5 seems relatively weak. Would it be possible to perform a FACS analysis of the same samples both at T=0 and at 16 hours, by subjecting a fraction of the reaction to FACS at the beginning of the reaction. This would provide a good baseline for identifying whether the liposome populations shift in the FACS measurement relative to T0.

Reply: The experiment suggested by the reviewer was shown in Fig. S19 (now S21) and discussed on page 9. The same samples were analyzed by FACS at different time points, from T=0 to 16 hours, and a clear shift of the data points could be measured over time with the DNAREP module (Fig. S21). Fluorescence intensity thresholds were defined at T=0 hour and the percentage of gated events in both dsGreen (DNA replication reporter) and mCherry (lipid synthesis reporter) channels were calculated. We think that the results show strong evidences of DNA replication in ~2-10% of liposomes containing the full *DNAREP-PLsyn* genome and in ~10-20% of the liposomes containing only the *DNAREP* module after 4-8 hours.

We confirmed this finding by performing a kinetics using confocal imaging (see comment below).

The data shown in Fig 2e should be supported by a statistical test to indicate whether or not there are indeed significant differences in DNA replication observed.

Reply: We performed a statistical test in Fig. 2e and Fig. 5f as described in the figure legends: "DNA concentration changes between 0 h and 16 h were assessed using a log-ratio paired *t*-test. Log-transformed ratios of 16-h to 0-h values were calculated for each replicate and each corresponding gene (*pssA* and *p2*). A one-sample *t*-test was then performed to determine if the mean log-ratio significantly differed from zero ($p < 0.05$). * $p \leq 0.05$, ** $p \leq 0.01$, and *** $p \leq 0.001$."

The differences in DNA concentrations between the 0-h and 16-h samples were statistically significant ($p < 0.05$). The *p* value is the lowest at 34 °C suggesting more efficient DNA replication. We have not run a statistical test to compare the amplification between the three temperatures.

Since this qPCR is a bulk assay it likely can't exclude the possibility that DNA replication, if it occurs, actually occurs inside liposomes or outside of liposomes (liposomes could burst and release their content and DNA replication could occur in the supernatant)?

Reply: We excluded the possibility that DNA replication and gene expression occurred outside liposomes by the external addition of DNase. Nonencapsulated DNA or DNA released from bursting liposomes would be degraded by DNase. We then inactivated DNase before qPCR measurements to allow for the quantification of the internal DNA released from bursting liposomes. Note that in qPCR experiments, the negative control sample is from T=0 hour. Therefore, the measured DNA concentration corresponds to the amount of input, encapsulated DNA.

Moreover, we confirmed that DNA replication occurred inside liposomes by confocal microscopy. When adding the intercalating dye dsGreen in liposome samples, no fluorescence signal was detected outside liposomes, only inside liposomes.

Direct imaging of liposomes as shown in Fig 3 is more convincing than the FACS data in regards to DNA replication, although here it would be good to provide a time-course as well (image liposomes at T=0 and then repeat imagine at reasonable intervals of every few hours). The following experiments in Fig 4 and 5 also nicely add to the evidence that DNA replication indeed occurs in liposomes. Could it therefore be that FACS is maybe not an ideal method for characterizing these liposomes as they could rupture and disintegrate in the FACS for example?

Reply: Thank you for suggesting to directly image liposomes over time. We performed a time-course experiment to complement the FACS data shown in Fig. S21. The results are provided in the new Supplementary Fig. 26. and confirm the increasing number of liposomes exhibiting replication separately or concurrently with lipid synthesis.

In general, liposome imaging is the method of choice as it provides direct observation and enables extraction of spatial features. On the other hand, FACS provides high-throughput analysis of liposome samples and its utilization is easier. We therefore decided to combine both techniques. We cannot rule out the possibility that samples get damaged on FACS fluidics or sample handling (filtering step). However, we expect this would similarly affect liposomes regardless of their activity states, and thus does not influence phenotypic mapping.

It remains somewhat curious why such a low fraction of liposomes are exhibiting DNA replication as well as liposome synthesis...

Reply: We carefully investigated this point in the revised manuscript and performed additional experiments that showed that a higher proportion of functional synthetic cells can be produced by improving the quality and purity of the template DNA.

In the Discussion section, we added: “We suspected that the robustness of the system may be limited by the DNA quality and purity. Using a stable source of DNA, namely a sequence-verified circular plasmid to generate the linear DNAREP-PLsyn template, improved the system’s efficiency over a factor of 3.5 with ~30% of fully functional synthetic cells. We attribute this improvement to a higher purity of the full-length DNA that also enables to work with increased concentrations of active template. The remaining fraction of nonfunctional synthetic cells probably arises from deficient molecular compositions inside vesicles, which cannot support all functions concurrently. For instance, differences in the loading of DNA or translation machinery between liposomes probably lead to varying levels of expressed genes, while uneven supply of substrates or cofactors for DNAREP and PLsyn would result in varying outputs of the modules. Even if it is clear that prototype synthetic cells will not have the level of control and robustness of natural living cells, a major challenge will be to mitigate this heterogeneity and increase the proportion of active vesicles.”

Moreover, in the Results section, line 331, we added a paragraph and included four new supplementary figures reporting these findings:

“High amounts of the PCR-amplified genome could be obtained, allowing for nanomolar input concentrations. Its expression in liposomes improved the robustness and efficiency of the system, yielding ~30% (vs 8%) of fully active synthetic cells and over 37% (vs 17%) with at least one functional module, while reducing variability between replicates (Supplementary Figs. 23-25). The change of phenotype map compared to Fig. 3c (template obtained by overlapping PCR) suggests an increased ability of DNAREP-active liposomes to also produce phospholipids.